# SpecAR-Net: Spectrogram Analysis and Representation Network for Time Series

## Abstract

Time series analysis has important applications in many fields. Representing temporal-structured samples is crucial for time series analysis tasks. Recently, several advanced deep learning models, *i.e.*, recurrent neural networks, convolutional neural networks, and transformer-style models, have been successively applied in temporal data representation, yielding notable results. However, most existing methods model and represent the variation patterns within time series solely in time domain. As a highly abstracted information entity, 1D time series data couples various patterns such as trends, seasonality, and dramatic changes (instantaneous high dynamic), it is difficult to exploit these highly coupled properties merely by analysis tools on purely time domain. To this end, we present **Spec**trum **A**nalysis and **R**epresentation **Net**work (SpecAR-Net). SpecAR-Net aims at learning more comprehensive representations by modeling raw time series in both time and frequency domain, where an efficient joint extraction of time-frequency features is achieved through a group of learnable 2D multi-scale parallel complex convolution blocks. Experimental results show that the SpecAR-Net achieves excellent performance on 5 major downstream tasks *i.e.*, classification, anomaly detection, imputation, long- and short-term series forecasting.

## 1 Introduction

With the advent of the era of "Internet of Things" and "Comprehensive Perception", various sensors have been extensively deployed and utilized, leading to an explosive growth in the scale of time series (Cook et al., 2020). Extracting valuable information from massive time series has become increasingly crucial. As a result, the time series analysis has attracted a growing number of researchers. Currently, time series data analysis has been widely applied in numerous fields, *e.g.*, finance (Livieris et al., 2020), electricity (Cai et al., 2020), transportation (Gasparin et al., 2021), and the healthcare sector (Stoean et al., 2020), etc.

Recently, deep learning is playing a crucial role in time series analysis. With the powerful feature representation capability, many deep time series learning methods have been proposed, and has achieved great success in classification, anomaly detection, short/long-term forecasting, imputation and other related tasks. One typical category of these methods is based on recurrent neural networks (RNNs) (Wang et al., 2022; Yu et al., 2021), where the sequence modeling is completed by recursively encoding the first-order dependency between the preceding and subsequent elements. However, when modeling long-term sequences, it is easy to encounter gradient vanishing and explosion problems, and it is also difficult to enjoy the advantages of parallel processing (Hochreiter, 1998). Another typical category is the convolution-based methods (Aksan & Hilliges, 2019; Thill et al., 2021), which can easily process sequential data in parallel. However, limited by the computation mechanism of shared convolutions in the local receptive fields, convolution models are often insufficient to characterize the long-term relationships. To overcome those shortcomings of recurrent and convolutional networks, the Self-Attention (SA) based Transformer (Liu et al., 2022a;b) has been proposed. Transformer balances the long-term dependency encoding capability and the benefits of parallel computing, resulting in widely used in various sequence modeling tasks. However, time series data is coupled with multiple patterns, and the temporal dependencies captured by point-by-point representation and aggregation are often submerged (Wu et al., 2021).

Form our opinion, the most significant limitation of the above methods is that they still rely on pure-time-domain modeling mechanism. As a highly abstract information body, time series data is formed by the coupling of multiple components such as trend (overall envelope), periodicity (multiple frequency components), mutagenicity (high frequency components), etc. Considering such highly-coupled property, it is almost impossible for pure-time-domain learning models to achieve complete semantic representation from time series data. To overcome such limitation, FEDFormer (Zhou et al., 2022) introduces frequency-domain analysis, and carries out SA in the frequency-domain to enhance the representation of time series data, resulting excellent results in long-term forecasting. In addition, TimesNet (Wu et al., 2023) builds a general time series representation module in a frequency-guided manner. It constructs a 2D period-time space by folding the time series data according to the dominant periods and using multi-scale 2D convolution to capture the intra-period-and inter-period-dependency, which achieves wonderful results in mainstream tasks.

These works show that frequency information is effective for time series representation. While these methods are eventually based on the single analysis domain, *i.e.*, lacking a more comprehensive utilization of time-frequency information. In particular, TimesNet solely relies on the frequency information obtained through Fourier transform to select dominant periodic components from time series. The subsequent modeling process is still carried out in the time domain, utilizing the data repeatedly guided by significant periods. There is no explicit analysis or thorough utilization of the frequency information. In this paper, to further explore the potential value of frequency information in time series modeling while preserving the temporal variation characteristics, a unified time-frequency **spec**trogram **a**nalysis and **r**epresentation **net**work (SpecAR-Net) is constructed. SpecAR-Net aims at achieving joint analysis of time and frequency for time series. In order to overcome the limitations of semantic entanglement caused by the coupling of trend, periodicity and mutagenicity in time series data, several special treatments are incorporated in the model design:

i. For decoupling the periodicity characteristics from time series data, the time-frequency transform is used for better extraction of time-frequency variation patterns in a higher-dimensional feature space.

ii. The mutagenicity disrupt the stability of the semantic representation space for time series. To address such issue, a group of parallel multi-scale convolution blocks is designed to deeply explore the transient patterns.

iii. To capture the trend patterns, the order-preserving is added to the loss function. This learning strategy, guided by the "order" prior information, aims to capture the global trend patterns of the time series.

Overall, this approach has the following benefits: firstly, it overcomes the bottleneck of one-dimensional data representation by decoupling the multiple components of time series in a higher-dimensional data space. Secondly, a concise unified framework for learning cross-domain representation is constructed, which enables the joint analysis of time and frequency domain features in time series. Technically, to facilitate the universality of the proposed method, a plug-and-play SpecAR-Block is designed, which is compatible for most deep sequential models. The experimental results demonstrate that SpecAR-Net achieves good performance in five mainstream tasks, including classification, anomaly detection, long-term forecasting, short-term forecasting, and imputation. Our contributions are summarized in three folds:

i. A unified time-frequency joint representation framework is proposed. This framework decouples the features into three levels: global variation features (trend), local variation features (periodicity), and transient change features (mutagenicity), enabling more efficient deep semantic feature extraction for time series data.

ii. A plug-and-play time-series representation module, SpecAR-Block is proposed, which is compatible with various deep sequence modeling frameworks, *e.g.*, RNNs, CNNs and Tranformers. By utilizing time frequency transformation and 2D multi-scale parallel complex convolutions, it can generate comprehensive semantic representation for input sequence.

iii. A powerful deep sequential model called SpecAR-Net with strong generalization ability is designed. SpecAR-Net has exhibited strong performance across a range of widely-used time series analysis tasks, such as anomaly detection, classification, long/short-term prediction and missing value imputation.

## 2 RELATED WORK

In essence, SpecAR-Net is a deep sequence modeling or encoding method. Initially, most of these methods were based on multi-layer perceptrons (MLP). For example, an extended MLP for predicting exchange rate trends using interval time series is presented in (Maté & Jimeńez, 2021). LightTS (Zhang et al., 2022) introduced a fine-grained down-sampling strategy into an MLP and achieved excellent performance in long-term forecasting tasks. DLinear (Zeng, 2023) decomposed time series into trend and residual sequences and utilized two MLPs to model these sequences for forecasting tasks.

Then, as a method specifically used for time series data modeling, RNN was widely investigated. It utilizes a chain-like structure to simulate the dynamic behavior of time series, which helps extract temporal characteristics. Such as the long short-term memory (LSTM) model used in (Hochreiter & Schmidhuber, 1997). And LSTNet proposed in (Lai et al., 2018a), which utilizes both CNNs and RNNs to extract short-term local dependencies between variables and explore long-term patterns in time series trends, respectively. More recently, LSSL (Albert Gu & Re., 2022) achieved effective modeling of long time series by parameterizing the continuous-time, recurrent, and convolutional views of the state space model.

Admittedly, RNNs are a naturally suitable model for dealing with time series. However the risk of gradient vanishing/explosion and limitation of serial computing have the obstacles for RNNs. In this context, CNNs are also favored. For example, dilated convolutions were utilized as an encoder to accept variable-length inputs for time series data modeling (Bai et al., 2018). TCN (Franceschi et al., 2019) employs multiple 1D convolutions to extract temporal information across different scales of feature maps, demonstrating certain advantages in extracting deep semantic features from time series. There's also research that indicates that CNNs exhibit superior performance to RNNs in time series modeling (Chen & Shi, 2021).

In recent years, Transformers have shown remarkable performance in the field of time series modeling (Nikita Kitaev & Levskaya, 2020). By utilizing SA mechanisms, these methods possess inherent network architecture advantages in capturing temporal dependencies in time series. As a result, they have become popular approaches in the field of time series analysis. For instance, informer (Zhou et al., 2021a) design ProbSparse SA mechanism and distillation operations to reduce the compuation complexity and memory consumption of the vanilla version. Inspired by the principle of exponential smoothing, ETSFormer (Woo et al., 2022) has been devised to improve the accuracy of time series data prediction by using novel Exponential Smoothing Attention (ESA) and Frequency Attention (FA) mechanisms.

These above methods provide many valuable ideas and practical tools for time series analysis. However, the modeling mechanisms are purely time-domain, which are difficult to describe and encode the highly coupled contents of the sequences, comprehensively. Considering such limitation, frequency information were incorporated into the deep models, which have achieved promising results, *e.g.*, FEDformer (Zhou et al., 2022) and TimesNet (Wu et al., 2023).

Although both of FEDformer and TimesNet emphasize the use of frequency information, they are essentially single-dimensional modeling mechanisms in terms of processing process, *i.e.*, while Fedformer relies exclusively on attention mechanisms in the frequency domain, TimesNet primarily utilizes time-domain modeling guided by significant frequency information. Hence, there still lacks of an effective time-frequency joint deep modeling method for time series analysis. Motivated by this, this paper specifically emphasizes modeling and analyzing time series data in the time-frequency domain, and attempts to provide a unified framework for a variety of mainstream tasks.

## 3 METHODOLOGY

In order to establish a comprehensive unified representation for time series, this paper proposes SpecAR-Net from the perspective of joint time-frequency analysis. Firstly, Short Time Fourier Transform(STFT) is used to map the time series data from the time domain into the time-frequency domain, resulting in a transform of data structure from 1D to 2D data space. Then, a group of multi-scale parallel complex convolution blocks, which efficiently extracts and fuses time-frequency

characteristics of the time series data. Through this process, we achieve a unified representation of the time series data in both time and frequency domains.

## 3.1 SPECAR-BLOCK

As shown in 1, the backbone of SpecAR-Net is composed of several stacking SpecAR-Blocks. Concretely, given one time series sample, $\mathbf{X} \in \mathbb{R}^{T \times N}$, where $T$ and $N$ denotes time length and data dimension, respectively. A high-dimensional mapping of $\mathbf{X}$ is performed at the very beginning as:

$$\mathbf{X}^0 = \text{Embed}(\mathbf{X}) \tag{1}$$

Where $\mathbf{X}^0 \in \mathbb{R}^{T \times M}$ is the encoded features generated by the embedding layer $\text{Embed}(\cdot) : \mathbb{R}^N \to \mathbb{R}^M$, which consists of three components: position embedding, global time stamp embedding, and scalar projection.

Then for the SpecAR-Net with $L$ blocks, the $l$-th ($l = 1, \ldots, L$) layer can be formalized as:

$$\mathbf{X}^l = \text{SpecAR}(\mathbf{X}^{l-1}) + \mathbf{X}^{l-1} \tag{2}$$

Where $\text{SpecAR}(\cdot) : \mathbb{R}^{T \times M} \to \mathbb{R}^{T \times M}$ denotes the SpecAR time-frequency encoding process, the output $\mathbf{X}^l$ is calculated by the SpecAR encoding along with a short-cut connection of $l-1$-th layer.

As can be seen from the detailed part of SpecAR-Block on the right of figure 1, each block consists of three core modules: time-frequency transformation, multi-scale complex convolutions, and feature aggregation. In specific, the time-frequency transformation (TFT) is performed to convert the temporal input features $\mathbf{X}^{l-1}$ into time-frequency structured (*i.e.*, spectrogram) complex tensor, $\mathbf{S}^{l-1} \in \mathbb{C}^{M \times T \times F}$, where $F$ denotes the number of frequency bins. Then a group of parallel multi-scale 2D complex convolutions is used to encode the complex tensor. This process can be formalized as follows:

$$\mathbf{S}^{l-1} = \text{TFT}(\mathbf{X}^{l-1})$$
$$\mathbf{S}^{(l-1)^\dagger} = \text{MS-Conv}^\dagger(\mathbf{S}^{l-1}) \tag{3}$$

Where $\text{TFT}(\cdot) : \mathbb{R}^T \to \mathbb{R}^{T \times F}$ denotes a dimension/channel-parallel time-frequency operator, which can be fulfilled by STFT (by default in this paper) or Wavelet Transform (WT). And $\text{MS-Conv}^\dagger(\cdot)$ denotes the parallel-computed complex convolutions with different dilation rates (sampling rates) in time-frequency receptive field of $\mathbf{S}^{l-1}$. Assume we have $K$ different convolution blocks, then the output tensor will be in the form of $\mathbf{S}^{(l-1)^\dagger} \in \mathbb{C}^{M \times T \times F \times K}$. More details of TFT and paralleled multi-scale complex convolutions are in Sec. 3.2 and 3.3.

Finally, for the feature aggregation stage, a block-wise average pooling is first conducted to compress the stacked feature tensor, $\mathbf{S}^{(l-1)^\dagger}$ obtained from $\text{MS-Conv}^\dagger(\cdot)$. Then a linear projection is used to transform the complex compressed features as real ones. This stage can be formalized as:

$$\mathbf{X}^{l\dagger} = \text{Linear} \left\{ \text{Re} \left[ \text{Avg}^{\text{Blk}} \left( \mathbf{S}^{(l-1)^\dagger} \right) \right], \text{Im} \left[ \text{Avg}^{\text{Blk}} \left( \mathbf{S}^{(l-1)^\dagger} \right) \right] \right\} \tag{4}$$

Where, $\text{Avg}^{\text{Blk}}(\cdot) : \mathbb{C}^K \to \mathbb{C}$ denotes the block-wise average pooling, $\text{Re}/\text{Im}[\cdot]$ is the element-wise real/complex part extractor, and $\text{Linear}(\cdot)$ denotes the complex-to-real linear projection. To further utilize the advantage of the skip connection, the fused time-frequency features, $\mathbf{X}^{l\dagger} \in \mathbb{R}^{M \times T \times F}$ will be average-pooled along the frequency domain and transposed to get the shape-compatible output tensor in $\mathbb{R}^{T \times M}$.

## 3.2 TIME-FREQUENCY TRANSFORMATION

In order to decouple and analyze the periodic characteristics of time series data while maintaining its temporal structure, we incorporate TFT in our SpecAR-Block. TFT can be fulfilled by the classic STFT or WT, which facilitates more efficient joint extraction of time-frequency features using 2D convolutions in subsequent learning stages. Following simplicity design principle, we use STFT by default. Fig. 2 illustrates the process of TFT for a given time series sample in an intuitive way.

Following the symbol definition above, for each channel of the given input sequence $\mathbf{X}^l \in \mathbb{R}^{T \times M}$ for $l + 1$-th SpecAR-Block, the TFT calculation is formalized as:

$$\mathbf{S}^l_m[t, f] = \sum_{\tau=t-n}^{t+n} \mathbf{X}^l_m[\tau] h(\tau - t) e^{-j2\pi f\tau} \tag{5}$$

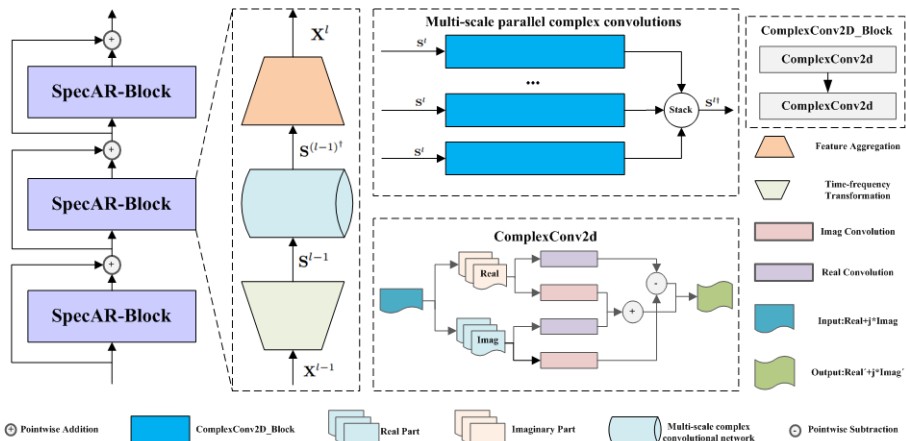

Figure 1: The overall framework of our approach.

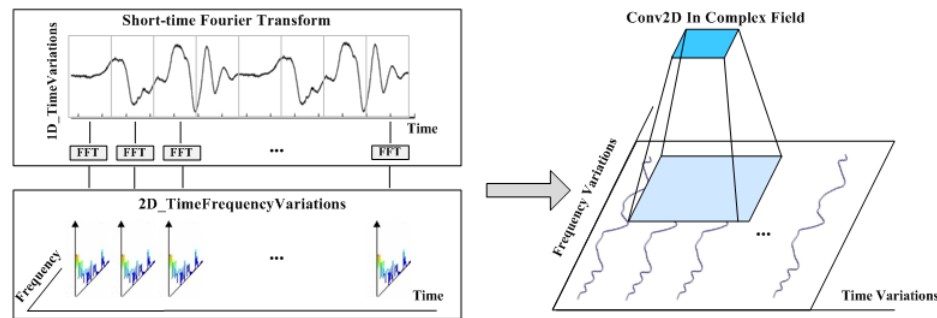

Figure 2: Time-frequency transform.

Where $\mathbf{S}_m^l \in \mathbb{C}^{T \times F}$ is the discrete STFT results, *i.e.*, spectrogram of $m$-th channel input sequence $\mathbf{X}_m^l \in \mathbb{R}^{T \times 1}$. Both window size and number of sample points for FFT are $2n+1$, the hop length is set as 1 so that the input and output can have the same time length. And the commonly used Hamming window function $h(t) = 0.54 - 0.46 \cos\left(\frac{\pi t}{n}\right)$ is employed to modulate the input sequence.

After applying the TFT to the time series data, the highly coupled pure time domain data is expanded into a time-frequency representation. This transformation will enable the subsequent feature learning part to more intuitively analyze the periodic components of the input sequence and their respective evolution trends on the time-frequency distribution. However, it should be noted that the spectrogram actually carries not only the amplitude but also the phase information in each time-frequency unit, these are retained in the complex numbers. Hence, to fully utilize those contents, special treatment should be taken into consideration in the subsequent learning stage, which is discussed in the following section. Beyond that, as a crucial role in accurately representing frequency-domain features, window length determines the frequency resolution of the resulting spectrogram. Therefore, the effect of window length is also investigated in the experiments.

## 3.3 MULTI-SCALE COMPLEX CONVOLUTIONS

To make full use of the information in the spectrogram representation, *i.e.*, the phase and the amplitude, the complex convolutions are utilized in SpecAR-Net. In addition, multi-scale convolution kernels for parallel computation is introduced to alleviate the contradiction of time-frequency resolution in TFT stage.

In order to avoid introducing more learning parameters, we use different dilation rates to achieve multi-scale feature extraction. Then the small network is designed to be constructed with $K$ complex convolution blocks with different dilation rates but the same kernel size, $3 \times 3$. Given the input time-frequency tensor of $l$-th SpecAR-Block, $\mathbf{S}^l \in \mathbb{C}^{M \times T \times F}$, The forward computation process can be roughly expressed as:

$$\mathbf{S}^{l\dagger} = \texttt{Stack}\left(\{\texttt{Conv}_k^{\dagger}\left(\mathbf{S}^l; d[k]\right)\}_{k=1}^K\right) \qquad (6)$$

Where $\text{Conv}_k^\dagger \left( \cdot; d[k] \right) : \mathbb{C}^M \to \mathbb{C}^M$ denotes the $k$-th convolution block with dilation rate of $d[k] = 2k + 1$. The specific calculation process for each $\text{Conv}_k^\dagger$ is as follows: assuming a complex convolution kernel $\mathbf{w} = (\mathbf{a} + j \odot \mathbf{b})$, and a complex input tensor $\mathbf{h} = (\mathbf{c} + j \odot \mathbf{d})$, the complex convolution process, denoted as:

$$\mathbf{w} * \mathbf{h} = (\mathbf{a} + j \odot \mathbf{b}) * (\mathbf{c} + j \odot \mathbf{d}) = (\mathbf{a} * \mathbf{c} - \mathbf{b} * \mathbf{d}) + j \odot (\mathbf{a} * \mathbf{d} + \mathbf{b} * \mathbf{c}) \tag{7}$$

Finally, the ouput tensors of all the complex convolution blocks will be stacked together in block-wise to form the multi-scale feature tensor $\mathbf{S}^{l\dagger} \in \mathbb{C}^{M \times T \times F \times K}$.

## 3.4 TEMPORAL ORDER PRESERVING

To capture the global trend patterns of the input time series, the temporal order preserving (TOP) constraint is incorporated into our SpecAR-Net. This constraint is achieved by adding an order regression loss term on the basis of the original prediction loss. In practice, we use a temporal-shared learning function to construct such TOP loss term.

Given the final embeddings of SpecAR-Net, $\mathbf{X}^* \in \mathbb{R}^{T \times M}$ for an input sequence. The learning function $\Phi(\cdot; \mathbf{u}) : \mathbb{R}^M \to \mathbb{R}$ will encode each $\mathbf{X}^*[t]$ as:

$$\Phi\left(\mathbf{X}^*[t]; \mathbf{u}\right) \mapsto t \tag{8}$$

Where $\mathbf{u}$ is the temporal-shared learning parameters and $t = 1, \ldots, T$ is the time index of $\mathbf{X}^*[t]$. Based on this order regression mechanism, the TOP loss term for the current input sequence can be formalized as follows:

$$\mathcal{L}_{\text{TOP}} = \frac{\lambda}{2} \sum_{t=1}^{T} \|\Phi\left(\mathbf{X}^*[t]; \mathbf{u}\right) - t\|_2^2 + \mathcal{R}(\mathbf{u}) \tag{9}$$

Where $\mathcal{L}_{\text{TOP}}$ denotes the TOP loss term, $\lambda$ is the order regression penalty factor and $\mathcal{R}(\mathbf{u})$ is the regularization term for $\mathbf{u}$. Then the final loss can be expressed as the weighted summation of $\mathcal{L}_{\text{TOP}}$ and the original loss for the current learning task, **e.g.**, mean square error for forecasting or cross-entropy for classification.

## 4 EXPERIMENTS

To verify the effectiveness and superiority of SpecAR-Net, a comprehensive set of experiments is conducted over 5 mainstream tasks, *i.e.*, classification, anomaly detection, long/short-term forecasting and imputation. The benchmark datasets and corresponding experimental configurations are shown in Tab. 1:

Table 1: The experiments configurations.

| No | Tasks | Datasets | Metrics | Series Length |
|---|---|---|---|---|
| 1 | Forecasting | Long-term:ETT(4subsets),Electricity, Weather,Exchange,ILI | MSE,MAE | 6∼720 (ILI:24∼60) |
| | | Short-term:M4(6 subsets) | SMAPE,MASE,OWA | 6∼48 |
| 2 | Imputation | ETT(4 subsets),Electricity,Weather | MSE,MAE | 96 |
| 3 | Classification | UEA(10 subsets) | Accuracy | 29∼1751 |
| 4 | Anomaly Detection | SMD,MSL,SMAP,SWaT,PSM | Precision,Recall,F1-Socre | 100 |

The backbones of the compared state-of-the-art (SoTA) models including RNNs, CNNs, MLPs and Transformers. The details are as follows Tab. 2:

Table 2: The contrast models

| No. | Models | Details |
|---|---|---|
| 1 | MLP-based | LightTS,DLinear |
| 2 | RNN-based | LSTM,LSTNet,LSSL |
| 3 | CNN-based | TCN, TimesNet |
| 4 | Transformer-based | Autoformer(Wu et al. (2021)),FEDformer,Reformer(Nikita Kitaev & Levskaya (2020)), Pyraformer(Liu et al. (2022a)),Non-stationary Transformer(Liu et al. (2022b)), Informer,ETSformer |

Furthermore, for specific tasks, cutting-edge models are also mentioned in the SoTA comparison this experiment. Specifically, N-HiTS (Challu et al., 2023) and N-BEATS (Oreshkin et al., 2019) are compared in short-term forecasting. Transformer (Xu et al., 2022) is selected for comparison in anomaly detection. For classification, Rocket (Dempster et al., 2020) and Flowformer (Wu et al., 2022) are compared.

### 4.1 MAIN RESULT

Compared to other baseline methods, SpecAR-Net has achieved the best performance across all 5 tasks, as shown in Tab. 3 (where red and blue font denote the best and second-best results, respectively). Additionally, the results further validate the exceptional generalization ability of SpecAR-Net, which can be regarded as a unified framework in time series analysis.

Table 3: the comparison of model performance

| Models | SpecAR-Net (Ours) | TimesNet (2023) | Dlinear (2023) | ETSformer (2022) | LightTS (2022) | Stationary (2022) | FEDformer (2022) | Informer (2021) | Autoforer (2021) | Reformer (2020) |
|---|---|---|---|---|---|---|---|---|---|---|
| Classification(Accuracy) | **74.7** | **73.6** | 67.5 | 71.0 | 70.4 | 72.7 | 70.7 | 72.1 | 71.1 | 71.5 |
| Anomaly Detection(F1-Scores) | **86.45** | **86.34** | 82.46 | 82.87 | 84.23 | 82.08 | 84.97 | 78.83 | 84.26 | 77.31 |
| Short-term Forecasting(OWA) | **0.850** | **0.851** | 1.051 | 1.172 | 1.051 | 0.930 | 0.918 | 1.230 | 0.939 | 1.775 |
| Long-term Forecasting(MSE)(ILL) | **2..051** | 2.139 | 2.616 | 2.497 | 7.382 | **2.077** | 2.847 | 5.137 | 3.006 | 4.724 |
| Imputation(MSE)(ETTh1) | **0.071** | **0.078** | 0.201 | 0.202 | 0.284 | 0.094 | 0.117 | 0.161 | 0.103 | 0.122 |

### 4.2 CLASSIFICATION

Time series classification task can intuitively show the performance of our method in terms of high-level semantic representation of time series. The data used in this experiment is sourced from the UAE dataset (Bagnall et al., 2018), comprising ten sub-datasets that encompass practical tasks such as gesture recognition, action recognition, audio recognition, and medical diagnosis.

As shown in Fig. 3, SpecAR-Net has achieved remarkable results in classification task, with an average classification accuracy of 74.7%, surpassing other SoTA methods such as TimesNet (73.6%) and Flowformer (73%). It is worth noting that, compared to SpecAR-Net, TimesNet exhibits lower classification accuracy on most datasets, with an average accuracy reduction of 1.1%. Efficient feature extraction can be carried out simultaneously in both time-domain and frequency-domain by SpecAR-Net, facilitating the capture of higher-level semantic representations.

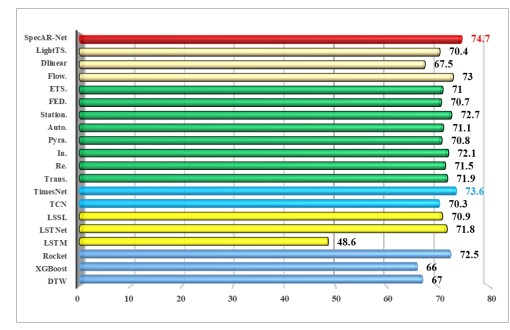

Figure 3: The Result of classification.

### 4.3 ANOMALY DETECTION

Anomaly detection plays a vital role in ensuring the orderly and secure operation of industrial production. However, anomaly detection often requires capturing exceptional signal within big data, which can easily get overwhelmed, making the detection task highly challenging. To fully validate the performance is such task, 5 widely-used datasets are employed, *i.e.*, SMD (Su et al., 2019), MSL and SMAP (Hundman et al., 2018), SWaT (Mathur & Tippenhauer, 2016), and PSM (Abdulaal et al., 2021). These datasets cover various real-world industrial applications, including service monitoring, spatial and earth sensing, and water treatment.

The results are presented in Tab. 4. It is evident that our method achieved the optimal performance in the anomaly detection task, outperforming other comparative methods. The advanced Transformer-based approaches, FEDformer and Autoformer, have also both achieved good performance(84.97% and 84.26%). Frequency-domain information is introduced into the attention mechanism of both models, further highlighting the effectiveness of frequency-domain information in time series representation. In comparison, our method enables the joint extraction of deep-level time-frequency features from both the time- and frequency-domains, thereby facilitating the capture of abnormal patterns existing in time series.

Table 4: The Result of Anomaly Detection

| Model Datasets | SpecAR-Net (ours) | TimesNet (ResNeXt) | TimesNet (Inception) | ETSformer (2022) | LightTS (2022) | Stationary (2022a) | FEDformer (2022) | Dlinear (2023) | LSSL (2022) | Informer (2021) | Anomaly (2021) | Pyraformer (2021a) |
|---|---|---|---|---|---|---|---|---|---|---|---|---|
| SMD | **86.55** | **85.81** | 85.12 | 83.13 | 82.53 | 84.62 | 85.08 | 77.1 | 71.31 | 81.65 | 85.49 | 83.04 |
| MSL | 81.72 | **85.15** | 84.18 | **85.03** | 78.95 | 77.5 | 78.57 | 84.88 | 82.53 | 84.06 | 83.31 | 84.86 |
| SMAP | **73.28** | **71.52** | 70.85 | 69.5 | 69.21 | 71.09 | 70.76 | 69.26 | 66.90 | 69.92 | 71.18 | 71.09 |
| SWaT | **93.42** | 91.74 | 92.10 | 84.91 | 79.88 | 93.19 | 87.52 | 85.76 | 81.43 | 83.10 | 91.78 | |
| PSM | 97.28 | **97.47** | 95.21 | 91.76 | 97.15 | **97.29** | 97.23 | 93.55 | 77.20 | 77.10 | 79.40 | 82.08 |
| Avg F1 | **86.45** | **86.34** | 85.49 | 82.87 | 84.23 | 82.08 | 84.97 | 82.46 | 76.74 | 78.83 | 80.50 | 82.57 |

## 4.4 LONG- AND SHORT-TERM FORECASTING

In the long-term forecasting task, a set of benchmark datasets were utilized, including ETT (Zhou et al., 2021b), Electricity (UCI), Traffic (PeMS), Weather (Wetterstation), Exchange (Lai et al., 2018b), and ILI (CDC), which cover the application demands of five major real-world scenarios. Each dataset contains a segment of continuous time series data, and sample data are obtained from these datasets using a sliding window approach. In the experiments, the input past length was set to 96, with ILL for 36. The prediction lengths is $[96, 192, 336, 720]$, with ILI for $[24, 36, 48, 60]$. In the short-term forecasting task, we utilized the M4 dataset (Makridakis et al., 2018), which comprises 100,000 time series . These data were collected at different sampling rates, including yearly, quarterly, monthly, weekly, daily, and hourly intervals, covering a wide range of domains such as finance, industry, and demographics. For our experiments,the prediction sequence lengths is $[6, 8, 13, 16, 24, 48]$.

Especially, all the results are averaged from four different prediction lengths for long-term forecasting, the results of short-term forecasting tasks are calculated as weighted averages from multiple datasets with varying sample intervals. The experiments are conducted in two rounds in total. In the first round, MSE is used as loss function, and it achieved good results in both short-term and long-term forecasting tasks, although it did not reach the optimal level. See Tab.7 and 8 in Appendix for more details. In the second round, SpecAR-Net was conducted by introducing a order-preserving into the loss function. As shown in Tab. 5 and 6, Our method achieves the best performance in both long- and short-term forecasting, indicating a positive role of the "order" information in time series forecasting. The order-preserving is equivalent to using the "order" information as prior knowledge to constrain the learning process of the model and compensate for the lost "order" information during feature extraction, ensuring that the model output possess a certain degree of sequentiality. At the same time, it also shows that our method is highly scalable.

Table 5: Short-term forecasting task (order-preserving). See Table 9 in Appendix for the full results.

| Models | SpecAR-Net (ours) | TimesNet (2023) | N-HiTS (2022) | N-BEATS (2019) | ETSformer (2022) | LightTS (2022) | Dlinear (2023) | FEDformer (2022) | Stationary (2022a) | Autoformer (2021) | Pyraformer (2021a) |
|---|---|---|---|---|---|---|---|---|---|---|---|
| SMAPE | **11.844** | **11.829** | 11.927 | 11.851 | 14.718 | 13.525 | 13.639 | 12.840 | 12.780 | 12.909 | 16.987 |
| MASE | **1.582** | **1.585** | 1.613 | 1.599 | 2.408 | 2.111 | 2.095 | 1.701 | 1.756 | 1.771 | 3.265 |
| OWA | **0.850** | **0.851** | 0.861 | 0.855 | 1.172 | 1.051 | 1.051 | 0.918 | 0.930 | 0.939 | 1.480 |

Table 6: Long-term forecasting task (order-preserving). See Table 10 in Appendix for the full results.

| Models | SpecAR-Net (ours) | | TimesNet (2023) | | ETSformer (2022) | | LightTS (2022) | | Dlinear (2023) | | FEDformer (2022) | | Stationary (2022a) | |
|---|---|---|---|---|---|---|---|---|---|---|---|---|---|---|
| Metrics | MSE | MAE | MSE | MAE | MSE | MAE | MSE | MAE | MSE | MAE | MSE | MAE | MSE | MAE |
| ETTm1 | **0.398** | **0.407** | **0.400** | **0.406** | 0.429 | 0.425 | 0.435 | 0.437 | 0.403 | 0.407 | 0.448 | 0.452 | 0.481 | 0.456 |
| ETTm2 | **0.291** | **0.332** | **0.291** | **0.333** | **0.293** | 0.342 | 0.409 | 0.436 | 0.35 | 0.401 | 0.305 | 0.349 | 0.306 | 0.347 |
| ETTh1 | 0.458 | 0.455 | 0.458 | **0.450** | 0.542 | 0.510 | 0.491 | 0.479 | **0.456** | **0.452** | **0.440** | 0.460 | 0.570 | 0.537 |
| ETTh2 | **0.416** | **0.427** | **0.414** | **0.427** | 0.439 | 0.452 | 0.602 | 0.543 | 0.559 | 0.515 | 0.437 | **0.449** | 0.526 | 0.516 |
| Eelctricity | **0.192** | **0.294** | **0.192** | 0.295 | 0.208 | 0.323 | 0.229 | 0.329 | 0.212 | 0.300 | 0.214 | 0.327 | **0.193** | 0.296 |
| Traffic | 0.625 | **0.335** | **0.620** | **0.336** | 0.621 | 0.396 | 0.622 | 0.392 | 0.625 | 0.383 | **0.610** | 0.376 | 0.624 | 0.340 |
| Weather | 0.257 | 0.284 | 0.259 | 0.287 | 0.271 | 0.334 | 0.261 | 0.312 | 0.265 | 0.317 | 0.309 | 0.360 | 0.288 | 0.314 |
| ExchangeRate | **0.384** | **0.425** | 0.416 | 0.443 | 0.410 | 0.427 | 0.385 | 0.447 | **0.354** | **0.414** | 0.519 | 0.500 | 0.461 | 0.454 |
| ILL | **2.051** | **0.903** | 2.139 | 0.931 | 2.497 | 1.004 | 7.382 | 2.003 | 2.616 | 1.090 | 2.847 | 1.144 | **2.077** | **0.914** |

## 4.5 IMPUTATION

The imputation task primarily relies on historical data to recover the missing data. This technique serves as the foundation of big data analytics, ensuring the temporal and spatial integrity of time series, thus supporting various subsequent tasks such as forecasting, classification, and anomaly detection. This experiment was conducted on 6 benchmark datasets, including ETT (w/ 4 subsets), Electricity and Weather. Random masking with masking rates of $[12.5\%, 25\%, 37.5\%, 50\%]$ was employed to simulate missing values.

The experiment was conducted in two rounds. The first round of the experiment was conducted without order-preserving. And SpecAR-Net exhibits consistent performance with TimesNet, which is the best-performing method among the comparison methods. See Tab.11 in Appendix for more details. Tab. 7, which presents the experimental results after incorporating order-preserving, shows that SpecAR-Net achieves the best performance. So, This indicates that the monotonicity constraint is beneficial for capturing the global trend patterns in time series. Furthermore, it also suggests that SpecAR-Net possesses strong capabilities in extracting time- and frequency-varying patterns.

## 4.6 DETAILED ANALYSIS

Table 7: Imputation tasks(order-preserving). See Table 12 in Appendix for the full results.

| Models | SpecAR-Net (ours) | | TimesNet (2023) | | ETS. (2022) | | LightTS (2022) | | DLinear (2023) | | FED. (2022) | | Stationary (2022a) | | Auto. (2021) | | Pyra. (2021a) | | In. (2021) | |
|---|---|---|---|---|---|---|---|---|---|---|---|---|---|---|---|---|---|---|---|---|
| Metrics | MSE | MAE | MSE | MAE | MSE | MAE | MSE | MAE | MSE | MAE | MSE | MAE | MSE | MAE | MSE | MAE | MSE | MAE | MSE | MAE |
| ETTm1 | **0.026** | **0.105** | 0.027 | 0.107 | 0.12 | 0.253 | 0.104 | 0.218 | 0.093 | 0.206 | 0.062 | 0.177 | 0.036 | 0.126 | 0.051 | 0.15 | 0.717 | 0.57 | 0.071 | 0.188 |
| ETTm2 | 0.21 | **0.087** | **0.022** | **0.089** | 0.208 | 0.327 | 0.046 | 0.151 | 0.096 | 0.208 | 0.101 | 0.215 | 0.026 | 0.099 | 0.029 | 0.105 | 0.465 | 0.508 | 0.156 | 0.292 |
| ETTh1 | **0.071** | **0.178** | 0.078 | 0.187 | 0.202 | 0.329 | 0.284 | 0.373 | 0.201 | 0.306 | 0.117 | 0.246 | 0.094 | 0.201 | 0.103 | 0.214 | 0.842 | 0.682 | 0.161 | 0.279 |
| ETTh2 | **0.046** | **0.141** | 0.049 | 0.146 | 0.367 | 0.436 | 0.119 | 0.25 | 0.142 | 0.259 | 0.163 | 0.279 | 0.053 | 0.152 | 0.055 | 0.156 | 1.079 | 0.792 | 0.337 | 0.452 |
| Electricity | **0.092** | **0.210** | 0.092 | 0.210 | 0.214 | 0.339 | 0.131 | 0.262 | 0.132 | 0.26 | 0.13 | 0.259 | **0.100** | **0.218** | 0.101 | 0.225 | 0.297 | 0.382 | 0.222 | 0.328 |
| Weather | **0.031** | **0.057** | 0.03 | **0.054** | 0.076 | 0.171 | 0.055 | 0.117 | 0.052 | 0.11 | 0.099 | 0.203 | 0.032 | 0.059 | **0.031** | **0.057** | 0.152 | 0.235 | 0.045 | 0.104 |

**Model Complexity & Performance**. To further analyze the performance of our method in the representation of time series data, we selected comparable models with better performance in classification and prediction tasks for model complexity analysis. Results in Fig. 4 show that better performance can be obtained by our method in

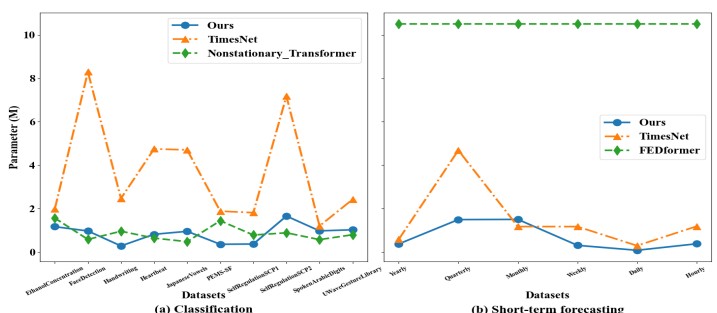

Figure 4: Model parameter scale.

the condition of less learnable parameters. This further illustrates the superiority of the proposed time-frequency joint learning model in this paper.

**Effects of TF Resolution**. The length window directly affects the time- and frequency-resolution of the time-frequency information obtained from STFT. The time- and frequency-resolution reflect the richness of information in the time and frequency domains, which has a significant impact on extracting time-frequency variation. Therefore, for this experiment, different window lengths $[4, 8, 16, 24, 48, 96, 192, 336]$ were selected to investigate their effects on the model performance. Figure 5 demonstrate that SpecAR-Net achieves optimal performance when the prediction lengths are $[96, 192, 336, 720]$, corresponding to window lengths with $[4, 24, 192, 192]$. This finding indicates that the requirements for time-frequency resolution vary across different

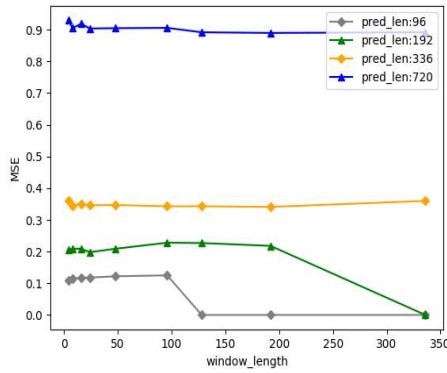

Figure 5: Effect of window length.

temporal analysis tasks, suggesting a varying dependency on both time- and frequency-features. According to the Heisenberg uncertainty principle (Mallet et al., 1999), it is impossible for the time- and frequency-resolution to simultaneously reach their optimal values. In order to ensure that our model has good time series representation capability while maintaining a suitable computational complexity, window lengths is [8, 16, 24] in this paper.

## 5 CONCLUSIONS

SpecAR-Net can be used as a universal foundational model for time-frequency representation and analysis of time series. Through the time-frequency transformation, SpecAR-Net overcomes the limitations of semantic representation in 1D time series caused by the coupling of multiple components such as trending, periodicity, and abruptness. This facilitates the simultaneous extraction and fusion of time-frequency variation patterns from a 2D space. Experimental results demonstrate that SpecAR-Net achieves optimal performance in 5 tasks, including classification, anomaly detection, long-term forecasting, short-term forecasting, and imputation.

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
