## A  APPENDIX

### A.1  IMPLEMENTATION-DETAILS

The code used for the benchmark method in this experiment is from TimesNets. Additionally, the code for SpecAR-Net is based on the modification of the TimesNet framework, primarily by replacing TimesBlock with SpecAR-Block. The deep learning framework used is PyTorch (version 1.13.1), and the GPU is two NVIDIA RTX 3090 Ti 24GB.

Model Hyperparameter Configuration: window functions: Hanning, Hamming, and rectangular windows. The window length, denoted as **win_len**, and the FFT window length, denoted as **n_fft**, are chosen from the range [8, 16, 24]. The **hop_length** represents the overlap between adjacent windows to prevent loss of temporal information. The results of the long-term forecasting(ETTh2) and classification(PEMS-SF) for different values of **hop_length** (predicting time series lengths: 96, 192, 336, 720) are illustrated in Figure 1. It can be observed that the impact of **hop_length** on the long-term forecasting tasks is relatively small. Additionally,as **hop_length** gradually increases, classification accuracy also tends to decrease. So, **hop_length** is 1 in this paper.

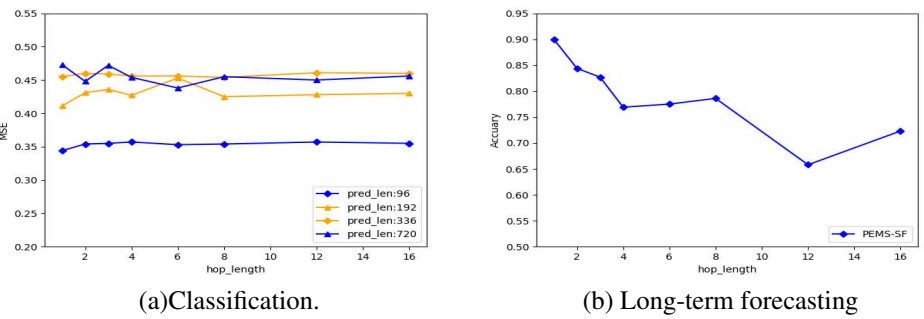

(a)Classification.                    (b) Long-term forecasting

Figure 1: Sensitivity analysis of the model to hop_length.

Parameters related to the complex-domain convolution network: **d_mode** and **d_ff** are selected from the range [16, 512]. **e_layers** denotes the number of SpecAR-Block, which ranges is [1,2,3,4, 5]. **conv_layers** represents the number of l $ComplexConv2D\_Block$, which chosen from the range [3, 6].

Metrics: In the classification task, accuracy is used as the metric. For anomaly detection tasks, the F1 score is utilized, which is the harmonic mean of precision(P) and recall(R). In the long- and short-term forecasting tasks, as well as the imputation task, the mean squared error (MSE) and the mean absolute error (MAE) are employed as metrics. In the short-term forecasting task, inspired by N-BEATS, the metrics used include the symmetric mean absolute percentage error (SMAPE), the mean absolute scaled error (MASE), and the overall weighted average (OWA). Notably, OWA is the measurement criterion utilized in the M4 competition. The formulas for calculating these respective metrics are presented as follows:

$$SMAPE = \frac{200}{H}\sum_{i=1}^{H}\frac{|X_i - \hat{X}_i|}{|X_i| + |\hat{X}_i|}, MAPE = \frac{100}{H}\sum_{i=1}^{H}\frac{|X_i - \hat{X}_i|}{|X_i| + |\hat{X}_i|} \tag{1}$$

$$MASE = \frac{1}{H}\sum_{i=1}^{H}\frac{|X_i - \hat{X}_i|}{\frac{1}{H-m}\sum_{j=m+1}^{H}|X_j - X_{j-m}|}, OWA = \frac{1}{2}[\frac{SMAPE}{SMAPE_{Na\text{ï}ve2}} + \frac{MASE}{MASE_{Na\text{ï}ve2}}] \tag{2}$$

Where represents the period of the time series data,**X**,$\hat{X}$ represent the original time series data and the corresponding predicted data, which the sequence length is **H** and the data dimension is **C**. **F** represents the data at the i-th future moment.

### A.2  COMPARED ANALYSIS WITH OTHER CONV-BASED NETWORKS

To validate the capability of the multi-scale parallel complex domain convolutional network, this experiment employed a dual-channel convolutional network and a feature encoding network $Embed(\cdot)$ as control methods. Three sets of experiments were conducted, namely anomaly detection, classification, and imputation. The experimental results are presented in Tables 1, 2, and 3, respectively. The control method employed in this experiment, namely the dual-channel convolutional network,

shares the same network architecture as the multi-scale parallel complex-domain convolutional network. However, it differs in the computation process by omitting the calculation of the correlation between the real and imaginary parts. The experimental results indicate that our proposed multi-scale parallel complex-domain convolutional network achieved the best performance in three sets of controlled experiments: anomaly detection, classification, and missing value imputation. Firstly, in comparison to $Embed(\cdot)$, our method demonstrated overwhelming advantages in all experiments, highlighting its effectiveness in handling time-frequency data. Secondly, our method consistently outperformed the dual-channel convolutional network in all controlled experiments, suggesting that the interplay between the real and imaginary parts, as designed in our approach, is more suitable for processing complex-frequency domain data, thereby enhancing the capability of extracting time-frequency patterns.

Table 1: SoTA Comparison in Imputation Task

| Backbone | | Complex Domain Convolutional Network(Ours) | | Dual Channel Convolutional Network | | Feature Encoding Network(Embed) | |
|---|---|---|---|---|---|---|---|
| Metric | | MSE | MAE | MSE | MAE | MSE | MAE |
| ETTm1 | 12.50% | **0.018** | **0.089** | **0.018** | **0.089** | 0.045 | 0.133 |
| | 25% | **0.022** | **0.098** | **0.022** | 0.099 | 0.07 | 0.165 |
| | 37.50% | 0.028 | 0.111 | **0.027** | **0.109** | 0.104 | 0.196 |
| | 50% | 0.035 | 0.124 | **0.034** | **0.122** | 0.14 | 0.229 |
| ETTm2 | 12.50% | **0.018** | 0.079 | **0.018** | **0.077** | 0.026 | 0.098 |
| | 25% | 0.020 | 0.084 | **0.019** | **0.083** | 0.033 | 0.114 |
| | 37.50% | **0.022** | 0.089 | **0.022** | **0.088** | 0.038 | 0.125 |
| | 50% | **0.025** | 0.097 | **0.025** | **0.096** | 0.045 | 0.137 |
| ETTh1 | 12.50% | **0.044** | **0.144** | 0.049 | 0.152 | 0.069 | 0.171 |
| | 25% | **0.061** | **0.169** | 0.063 | 0.171 | 0.102 | 0.208 |
| | 37.50% | **0.079** | **0.190** | 0.081 | 0.193 | 0.14 | 0.241 |
| | 50% | **0.098** | **0.210** | 0.108 | 0.22 | 0.186 | 0.277 |
| ETTh2 | 12.50% | 0.038 | 0.128 | **0.037** | **0.125** | 0.046 | 0.139 |
| | 25% | 0.042 | 0.136 | **0.041** | **0.134** | 0.056 | 0.156 |
| | 37.50% | 0.047 | 0.144 | **0.046** | **0.140** | 0.067 | 0.172 |
| | 50% | 0.056 | 0.157 | **0.053** | **0.151** | 0.079 | 0.187 |
| weather | 12.50% | **0.027** | **0.052** | **0.027** | 0.054 | 0.027 | 0.048 |
| | 25% | **0.028** | **0.052** | 0.029 | 0.057 | 0.031 | 0.059 |
| | 37.50% | **0.031** | **0.058** | 0.032 | 0.061 | 0.036 | 0.067 |
| | 50% | 0.036 | 0.066 | **0.035** | **0.064** | 0.042 | 0.075 |
| electricity | 12.50% | **0.086** | **0.202** | **0.086** | **0.202** | 0.086 | 0.205 |
| | 25% | **0.089** | **0.206** | 0.09 | 0.207 | 0.095 | 0.216 |
| | 37.50% | **0.094** | **0.212** | 0.095 | 0.213 | 0.104 | 0.227 |
| | 50% | **0.100** | **0.220** | **0.100** | 0.221 | 0.115 | 0.24 |
| $1^{st}$ Count | | 29 $1^{st}$ | | 29 $1^{st}$ | | | |

Table 2: Comparison of Different Feature Extraction Networks in Classification Task

| DataSets | Complex Domain Convolutional Network(Ours) | Dual channel convolutional network | Feature Encoding Network(Embed) |
|---|---|---|---|
| | Accuracy | | |
| EthanolConcentration | 0.327 | 0.281 | 0.27 |
| FaceDetection | 0.701 | 0.652 | 0.675 |
| Handwriting | 0.421 | 0.328 | 0.284 |
| Heartbeat | 0.78 | 0.746 | 0.756 |
| JapaneseVowels | 0.984 | 0.951 | 0.978 |
| PEMS-SF | 0.902 | 0.844 | 0.850 |
| SelfRegulationSCP1 | 0.922 | 0.891 | 0.925 |
| SelfRegulationSCP2 | 0.572 | 0.528 | 0.533 |
| SpokenArabicDigits | 0.995 | 0.994 | 0.975 |
| UWaveGestureLibrary | 0.869 | 0.647 | 0.856 |
| Average Accuracy | **0.7473** | **0.6862** | **0.7102** |

Table 3: Comparison of Different Feature Extraction Networks in Anomaly Detection Task

| DataSet | Complex Domain Convolutional Network | | | Dual channel convolutional network | | | Feature Encoding Network(Embed) | | |
|---|---|---|---|---|---|---|---|---|---|
| | Precision | Recall | F-score | Precision | Recall | F-score | Precision | Recall | F-score |
| SMD | 0.8874 | 0.8447 | 0.8655 | 0.8758 | 0.8104 | 0.8419 | 0.8671 | 0.7384 | 0.7976 |
| MSL | 0.8997 | 0.7487 | 0.8172 | 0.8777 | 0.7004 | 0.7791 | 0.7901 | 0.3707 | 0.5046 |
| SMAP | 0.8998 | 0.6181 | 0.7328 | 0.8997 | 0.555 | 0.6865 | 0.9011 | 0.5151 | 0.6555 |
| SWaT | 0.9155 | 0.9536 | 0.9342 | 0.9126 | 0.953 | 0.9324 | 0.9006 | 0.9559 | 0.9274 |
| PSM | 0.984 | 0.9619 | 0.9728 | 0.9854 | 0.9388 | 0.9615 | 0.9814 | 0.8375 | 0.9038 |
| Average F1 | **0.8645** | | | **0.8403** | | | **0.7578** | | |

## A.3 GENERALIZATION ABILITY

To verify the benefits of large-scale pretraining on model performance, this experiment aims to evaluate the performance of the model on a mixed dataset, which includes ETTh1, ETTh2, ETTm1, and ETTm2. It is important to note that ETTh1 and ETTh2 have an hourly sampling period, ETTm1 and ETTm2 have an the sampling period of 15 minutes. As a result, this mixed dataset contains more complex time- and frequency-variations, posing significant challenges in constructing effective time series representations. The experiment yielded results as shown in Table 4, indicating that our method achieved improved performance on all four sub-datasets through pre-training on the mixed

dataset. When compared to other methods, our approach outperformed them after pre-training, showcasing its superior feature extraction capability to enable effective handling of large-scale and complex datasets. Furthermore, our method demonstrated remarkable generalization and adaptability on the mixed dataset, implying its potential as a universal network framework for representing temporal data.

Table 4: Comparison between unified training and independent training for imputation task.

| Models | DataSets | | ETTm1 | | | | ETTm2 | | | | ETTh1 | | | | ETTh2 | | | |
|---|---|---|---|---|---|---|---|---|---|---|---|---|---|---|---|---|---|---|
| | Mask Ratio | | 12.50% | 25% | 37.50% | 50% | 12.50% | 25% | 37.50% | 50% | 12.50% | 25% | 37.50% | 50% | 12.50% | 25% | 37.50% | 50% |
| SpecAR-Net | Unified | MSE | **0.017** | **0.210** | **0.027** | **0.033** | **0.017** | **0.019** | **0.021** | **0.024** | **0.033** | **0.045** | **0.057** | **0.072** | **0.030** | **0.034** | **0.039** | **0.045** |
| | | MAE | **0.086** | **0.096** | **0.107** | **0.119** | **0.074** | **0.080** | **0.086** | **0.093** | **0.122** | **0.143** | **0.161** | **0.181** | **0.107** | **0.116** | **0.126** | **0.136** |
| | Independent | MSE | **0.018** | **0.022** | **0.028** | **0.035** | **0.018** | **0.020** | **0.022** | **0.025** | 0.044 | 0.061 | 0.079 | 0.098 | 0.038 | 0.042 | 0.047 | 0.056 |
| | | MAE | **0.089** | **0.098** | **0.111** | 0.124 | 0.079 | **0.084** | **0.089** | **0.097** | 0.144 | 0.169 | 0.190 | 0.210 | 0.128 | 0.136 | 0.144 | 0.157 |
| TimesNet | Unified | MSE | 0.019 | 0.023 | 0.028 | 0.037 | 0.018 | 0.02 | 0.022 | 0.025 | **0.035** | **0.046** | **0.057** | **0.075** | **0.032** | **0.036** | **0.040** | **0.047** |
| | | MAE | 0.091 | 0.099 | 0.109 | **0.123** | **0.075** | 0.081 | 0.086 | 0.095 | **0.126** | **0.144** | **0.159** | **0.181** | **0.112** | **0.119** | **0.129** | **0.140** |
| | Independent | MSE | 0.019 | 0.023 | 0.029 | 0.037 | 0.02 | 0.02 | 0.023 | 0.026 | 0.057 | 0.069 | 0.084 | 0.102 | 0.04 | 0.046 | 0.052 | 0.06 |
| | | MAE | 0.092 | 0.101 | 0.111 | 0.124 | 0.08 | 0.085 | 0.091 | 0.098 | 0.159 | 0.178 | 0.196 | 0.215 | 0.13 | 0.141 | 0.151 | 0.162 |
| FEDformer | Unified | MSE | 0.041 | 0.057 | 0.073 | 0.099 | 0.06 | 0.089 | 0.125 | 0.172 | 0.077 | 0.101 | 0.13 | 0.164 | 0.087 | 0.125 | 0.161 | 0.214 |
| | | MAE | 0.143 | 0.169 | 0.192 | 0.224 | 0.166 | 0.205 | 0.244 | 0.287 | 0.196 | 0.228 | 0.258 | 0.289 | 0.204 | 0.246 | 0.283 | 0.326 |
| | Independent | MSE | 0.035 | 0.052 | 0.069 | 0.089 | 0.056 | 0.08 | 0.11 | 0.156 | 0.07 | 0.106 | 0.124 | 0.165 | 0.095 | 0.137 | 0.187 | 0.232 |
| | | MAE | 0.135 | 0.166 | 0.191 | 0.218 | 0.159 | 0.195 | 0.231 | 0.276 | 0.19 | 0.236 | 0.258 | 0.299 | 0.212 | 0.258 | 0.304 | 0.341 |
| Autoformer | Unified | MSE | 0.034 | 0.048 | 0.06 | 0.078 | 0.023 | 0.027 | 0.03 | 0.034 | 0.066 | 0.086 | 0.114 | 0.133 | 0.042 | 0.049 | 0.055 | 0.065 |
| | | MAE | 0.122 | 0.146 | 0.163 | 0.185 | 0.091 | 0.102 | 0.109 | 0.117 | 0.174 | 0.2 | 0.229 | 0.247 | 0.135 | 0.147 | 0.157 | 0.171 |
| | Independent | MSE | 0.034 | 0.046 | 0.057 | 0.067 | 0.023 | 0.026 | 0.03 | 0.035 | 0.074 | 0.09 | 0.109 | 0.137 | 0.044 | 0.05 | 0.06 | 0.068 |
| | | MAE | 0.124 | 0.144 | 0.161 | 0.174 | 0.092 | 0.101 | 0.108 | 0.119 | 0.182 | 0.203 | 0.222 | 0.248 | 0.138 | 0.149 | 0.163 | 0.173 |

## A.4 FULL RESULT

The complete results of the five data analysis tasks are as follows: Table 5 contains the results of the classification task. Table 6 contains the results of the anomaly detection task. Tables 7 and 9 contain the results of the short-term forecasting task. Tables 8 and 10 contain the results of the long-term forecasting task. And Tables 11 and 12 contain the results of imputation task. Additionally,the red font and blue font in the table represent the best and second-best results, respectively. *. in the Transformers indicates the name of *.former.

Table 5: The result of classification task. We report the classification accuracy(%) as the result.

| Models | Classical methods | | | RNN | | | TCN | | Transformer | | | | | | | | | MLP | | SpecAR-Net |
|---|---|---|---|---|---|---|---|---|---|---|---|---|---|---|---|---|---|---|---|---|
| | DTW | XGBoost | Rocket | LSTM | LSTNet | LSSL | TimesNet | TCN | Trans. | Re. | In. | Pyra. | Auto. | Station. | FED. | ETS. | Flow. | Dlinear | LightTS. | |
| | (1994) | (2016) | (2020) | (1997) | (2018) | (2022) | (2023) | (2019) | (2017) | (2020) | (2021) | (2021a) | (2021) | (2022a) | (2022) | (2022) | (2022) | (2023) | (2022) | (ours) |
| EthanolConcentration | 32.3 | 43.7 | 45.2 | 32.3 | 39.9 | 31.1 | 35.7 | 28.9 | 32.7 | 31.9 | 31.6 | 30.8 | 31.6 | 32.7 | 31.2 | 28.1 | 33.8 | 32.6 | 29.7 | 32.7 |
| FaceDetection | 52.9 | 63.3 | 64.7 | 57.7 | 65.7 | 66.7 | 68.6 | 52.8 | 67.3 | 68.6 | 67 | 65.7 | 68.4 | 68 | 66 | 66.3 | 67.6 | 68 | 67.5 | 70.1 |
| Handwriting | 28.6 | 15.8 | 58.8 | 15.2 | 25.8 | 24.6 | 32.1 | 53.3 | 32 | 27.4 | 32.8 | 29.4 | 36.7 | 31.6 | 28 | 32.5 | 33.8 | 27 | 26.1 | 42.1 |
| Heartbeat | 71.7 | 73.2 | 75.6 | 72.2 | 77.1 | 72.7 | 78 | 75.6 | 76.1 | 77.1 | 80.5 | 75.6 | 74.6 | 73.7 | 73.7 | 71.2 | 77.6 | 75.1 | 75.1 | 78 |
| JapaneseVowels | 94.9 | 86.5 | 96.2 | 79.7 | 98.1 | 98.4 | 98.4 | 98.9 | 98.7 | 97.8 | 98.9 | 98.4 | 96.2 | 99.2 | 98.4 | 95.9 | 98.9 | 96.2 | 96.2 | 98.4 |
| PEMS-SF | 71.1 | 98.3 | 75.1 | 39.9 | 86.7 | 86.1 | 89.6 | 68.8 | 82.1 | 82.7 | 81.5 | 83.2 | 82.7 | 87.3 | 80.9 | 86 | 83.8 | 75.1 | 88.4 | 90.2 |
| SelfRegulationSCP1 | 77.7 | 84.6 | 90.8 | 68.9 | 84 | 90.8 | 91.8 | 84.6 | 92.2 | 90.4 | 90.1 | 88.1 | 84 | 89.4 | 88.7 | 89.6 | 92.5 | 87.3 | 89.8 | 92.2 |
| SelfRegulationSCP2 | 53.9 | 48.9 | 53.3 | 46.6 | 52.8 | 52.2 | 57.2 | 55.6 | 53.9 | 56.7 | 53.3 | 53.3 | 50.6 | 57.2 | 54.4 | 55 | 56.1 | 50.5 | 51.1 | 57.2 |
| SpokenArabicDigits | 96.3 | 69.6 | 71.2 | 31.9 | 100 | 100 | 99 | 95.6 | 98.4 | 97 | 100 | 99.6 | 100 | 100 | 100 | 100 | 98.8 | 81.4 | 100 | 99.5 |
| UWaveGestureLibrary | 90.3 | 75.9 | 94.4 | 41.2 | 87.8 | 85.9 | 85.3 | 88.4 | 85.6 | 85.6 | 85.6 | 83.4 | 85.9 | 87.5 | 85.3 | 85 | 86.6 | 82.1 | 80.3 | 86.9 |
| AverageAccuracy | 67 | 66 | 72.5 | 48.6 | 71.8 | 70.9 | **73.6** | 70.3 | 71.9 | 71.5 | 72.1 | 70.8 | 71.1 | 72.7 | 70.7 | 71 | 73.0 | 67.5 | 70.4 | **74.7** |

Table 6: The result of anomaly detection task. The P, R and F1 represent the represent the precision ,recall and F1-score(%). A higher value of P, R and F1 indicates a better performance.

| Datasets | SMD | | | MSL | | | SMAP | | | SWaT | | | PSM | | | Avg F1 |
|---|---|---|---|---|---|---|---|---|---|---|---|---|---|---|---|---|
| Metrics | P | R | F1 | P | R | F1 | P | R | F1 | P | R | F1 | P | R | F1 | |
| LSTM(1997) | 78.52 | 65.47 | 71.41 | 78.04 | 86.22 | 81.93 | 91.06 | 57.49 | 70.48 | 78.06 | 91.72 | 84.34 | 69.24 | 99.53 | 81.67 | 77.97 |
| Transformer(2017) | 83.58 | 76.13 | 79.56 | 71.57 | 87.37 | 78.68 | 89.37 | 57.12 | 69.7 | 68.84 | 96.53 | 80.37 | 62.75 | 96.56 | 76.07 | 76.88 |
| LogTrans(2019) | 83.46 | 70.13 | 76.21 | 73.05 | 87.37 | 79.57 | 89.15 | 57.59 | 69.97 | 68.67 | 97.32 | 80.52 | 63.06 | 98 | 76.74 | 76.6 |
| TCN(2019) | 84.06 | 79.07 | 81.49 | 75.11 | 82.44 | 78.6 | 86.9 | 59.23 | 70.45 | 76.59 | 95.71 | 85.09 | 54.59 | 99.77 | 70.57 | 77.24 |
| Reformer(2020) | 82.58 | 69.24 | 75.32 | 85.51 | 83.31 | 84.4 | 90.91 | 57.44 | 70.4 | 72.5 | 96.53 | 82.8 | 59.93 | 95.38 | 73.61 | 77.31 |
| Informer(2021) | 86.6 | 77.23 | 81.65 | 81.77 | 86.48 | 84.06 | 90.11 | 57.13 | 69.92 | 70.29 | 96.75 | 81.43 | 64.27 | 96.33 | 77.1 | 78.83 |
| Anomaly*(2021) | 88.91 | 82.23 | 85.49 | 79.61 | 87.37 | 83.31 | 91.85 | 58.11 | 71.18 | 72.51 | 97.32 | 83.1 | 68.35 | 94.72 | 79.4 | 80.5 |
| Pyraformer(2021a) | 85.61 | 80.61 | 83.04 | 83.81 | 85.93 | 84.86 | 92.54 | 57.71 | 71.09 | 87.92 | 96 | 91.78 | 71.67 | 96.02 | 82.08 | 82.57 |
| Autoformer(2021) | 88.06 | 82.35 | 85.11 | 77.27 | 80.92 | 79.05 | 90.4 | 58.62 | 71.12 | 89.85 | 95.81 | 92.74 | 99.08 | 88.15 | 93.29 | 84.26 |
| LSSL(2022) | 78.51 | 65.32 | 71.31 | 77.55 | 88.18 | 82.53 | 89.43 | 53.43 | 66.9 | 79.05 | 93.72 | 85.76 | 66.02 | 92.93 | 77.2 | 76.74 |
| Stationary(2022a) | 88.33 | 81.21 | 84.62 | 68.55 | 89.14 | 77.5 | 89.37 | 59.02 | 71.09 | 68.03 | 96.75 | 79.88 | 97.82 | 96.76 | **97.29** | 82.08 |
| Dlinear(2023) | 83.62 | 71.52 | 77.1 | 84.34 | 85.42 | 84.88 | 92.32 | 55.41 | 69.26 | 80.91 | 95.3 | 87.52 | 98.28 | 89.26 | 93.55 | 82.46 |
| ETSformer(2022) | 87.44 | 79.23 | 83.13 | 85.13 | 84.93 | **85.03** | 92.25 | 55.75 | 69.5 | 90.02 | 80.36 | 84.91 | 99.31 | 85.28 | 91.76 | 82.87 |
| LightTS(2022) | 87.1 | 78.42 | 82.53 | 82.4 | 75.78 | 78.95 | 92.58 | 55.27 | 69.21 | 91.98 | 94.72 | **93.33** | 98.37 | 95.97 | 97.15 | 84.23 |
| FEDformer(2022) | 87.95 | 82.39 | 85.08 | 77.14 | 80.07 | 78.57 | 90.47 | 58.1 | 70.76 | 90.17 | 96.42 | 93.19 | 97.31 | 97.16 | 97.23 | 84.97 |
| TimesNet(Inception) | 87.76 | 82.63 | 85.12 | 82.97 | 85.42 | 84.18 | 91.5 | 57.8 | 70.85 | 88.31 | 96.24 | 92.1 | 98.22 | 92.21 | 95.21 | 85.49 |
| TimesNet(ResNeXt) | 88.66 | 83.14 | **85.81** | 83.92 | 86.42 | **85.15** | 92.52 | 58.29 | **71.52** | 86.76 | 97.32 | 91.74 | 98.19 | 96.76 | **97.47** | **86.34** |
| SpecAR-Net(ours) | 88.74 | 84.47 | **86.55** | 89.97 | 74.87 | 81.72 | 89.98 | 61.81 | **73.28** | 91.55 | 95.36 | **93.42** | 98.40 | 96.19 | 97.28 | **86.45** |

Table 7: The result of short-term forecasting task.

| Models | | SpecAR-Net (ours) | TimesNet (2023) | N-HiTS (2022) | N-BEATS (2019) | ETS. (2022) | LightTS (2022) | Dlinear (2023) | FED. (2022) | Stationary (2022a) | Auto. (2021) | Pyra. (2021a) | In. (2021) | LogTrans (2019) | Re. (2020) | LSTM (1997) | TCN (2019) | LSSL (2022) |
|---|---|---|---|---|---|---|---|---|---|---|---|---|---|---|---|---|---|---|
| Yearly | SMAPE | 13.417 | 13.387 | 13.418 | 13.436 | 18.009 | 14.247 | 16.965 | 13.728 | 13.717 | 13.974 | 15.53 | 14.727 | 17.107 | 16.169 | 176.040 | 14.920 | 61.675 |
| | MASE | 2.992 | 2.996 | 3.045 | 3.043 | 4.487 | 3.109 | 4.283 | 3.048 | 3.078 | 3.134 | 3.711 | 3.418 | 4.177 | 3.800 | 31.033 | 3.364 | 19.953 |
| | OWA | 0.787 | 0.786 | 0.793 | 0.794 | 1.115 | 0.827 | 1.058 | 0.803 | 0.807 | 0.822 | 0.942 | 0.881 | 1.049 | 0.973 | 9.29 | 0.880 | 4.397 |
| Quarterly | SMAPE | 10.248 | 10.100 | 10.202 | 10.124 | 13.376 | 11.364 | 12.145 | 10.792 | 10.958 | 11.338 | 15.449 | 11.360 | 13.207 | 13.313 | 172.808 | 11.122 | 65.999 |
| | MASE | 1.201 | 1.182 | 1.194 | 1.169 | 1.906 | 1.328 | 1.520 | 1.283 | 1.325 | 1.365 | 2.35 | 1.401 | 1.827 | 1.775 | 19.753 | 1.360 | 17.662 |
| | OWA | 0.903 | 0.890 | 0.899 | 0.886 | 1.302 | 1 | 1.106 | 0.958 | 0.981 | 1.012 | 1.558 | 1.027 | 1.266 | 1.252 | 15.049 | 1.001 | 9.436 |
| Monthly | SMAPE | 12.921 | 12.670 | 12.791 | 12.677 | 14.588 | 14.014 | 13.514 | 14.260 | 13.917 | 13.958 | 17.642 | 14.062 | 16.149 | 20.128 | 143.237 | 15.626 | 64.664 |
| | MASE | 0.955 | 0.933 | 0.969 | 0.937 | 1.368 | 1.053 | 1.037 | 1.102 | 1.097 | 1.103 | 1.913 | 1.141 | 1.660 | 2.614 | 16.551 | 1.274 | 16.245 |
| | OWA | 0.897 | 0.878 | 0.899 | 0.880 | 1.149 | 0.981 | 0.956 | 1.012 | 0.998 | 1.002 | 1.511 | 1.024 | 1.34 | 1.927 | 12.747 | 1.141 | 9.879 |
| Others | SMAPE | 4.872 | 4.891 | 5.061 | 4.925 | 7.267 | 15.880 | 6.709 | 4.954 | 6.302 | 5.485 | 24.786 | 24.460 | 23.236 | 32.491 | 186.282 | 7.186 | 121.844 |
| | MASE | 3.293 | 3.302 | 3.216 | 3.391 | 5.240 | 11.434 | 4.953 | 3.264 | 4.064 | 3.865 | 18.581 | 20.960 | 16.288 | 33.355 | 119.294 | 4.677 | 91.650 |
| | OWA | 1.032 | 1.035 | 1.040 | 1.053 | 1.591 | 3.474 | 1.487 | 1.036 | 1.304 | 1.187 | 5.538 | 5.879 | 5.013 | 8.679 | 38.411 | 1.494 | 27.273 |
| W-Average | SMAPE | 11.991 | 11.829 | 11.927 | 11.851 | 14.718 | 13.525 | 13.639 | 12.840 | 12.780 | 12.909 | 16.987 | 14.086 | 16.018 | 18.200 | 160.031 | 13.961 | 67.156 |
| | MASE | 1.600 | 1.585 | 1.613 | 1.599 | 2.408 | 2.111 | 2.095 | 1.701 | 1.756 | 1.771 | 3.265 | 2.718 | 3.010 | 4.223 | 25.788 | 1.945 | 21.208 |
| | OWA | 0.860 | 0.851 | 0.861 | 0.855 | 1.172 | 1.051 | 1.051 | 0.918 | 0.93 | 0.939 | 1.48 | 1.230 | 1.378 | 1.775 | 12.642 | 1.023 | 8.021 |

Table 8: The result of long-term forecasting task. $Avg$ is average from all four predicton lengths

| Models | | SpecAR-Net (ours) | | TimesNet (2023) | | ETSformer (2022) | | LightTS (2022) | | Dlinear (2023) | | FEDformer (2022) | | Stationary (2022a) | | Autoformer (2021) | | Pyraformer (2021a) | | Informer (2021) | | LogTrans (2019) | | Reformer (2020) | | LSSL (2022) | | LSTM (1997) | |
|---|---|---|---|---|---|---|---|---|---|---|---|---|---|---|---|---|---|---|---|---|---|---|---|---|---|---|---|---|---|
| Metrics | | MSE | MAE | MSE | MAE | MSE | MAE | MSE | MAE | MSE | MAE | MSE | MAE | MSE | MAE | MSE | MAE | MSE | MAE | MSE | MAE | MSE | MAE | MSE | MAE | MSE | MAE | MSE | MAE |
| ETTm1 | 96 | 0.378 | 0.397 | 0.338 | 0.375 | 0.375 | 0.398 | 0.374 | 0.4 | 0.345 | 0.372 | 0.379 | 0.419 | 0.386 | 0.398 | 0.505 | 0.475 | 0.543 | 0.51 | 0.672 | 0.571 | 0.6 | 0.546 | 0.538 | 0.528 | 0.45 | 0.477 | 0.863 | 0.664 |
| | 192 | 0.425 | 0.419 | 0.374 | 0.387 | 0.408 | 0.41 | 0.4 | 0.407 | 0.38 | 0.389 | 0.426 | 0.441 | 0.459 | 0.444 | 0.553 | 0.496 | 0.557 | 0.537 | 0.795 | 0.669 | 0.84 | 0.7 | 0.658 | 0.592 | 0.469 | 0.481 | 1.113 | 0.776 |
| | 336 | 0.413 | 0.417 | 0.41 | 0.411 | 0.435 | 0.428 | 0.438 | 0.438 | 0.413 | 0.413 | 0.445 | 0.459 | 0.495 | 0.464 | 0.621 | 0.537 | 0.754 | 0.655 | 1.212 | 0.871 | 1.12 | 0.832 | 0.898 | 0.721 | 0.583 | 0.574 | 1.267 | 0.832 |
| | 720 | 0.499 | 0.466 | 0.478 | 0.450 | 0.499 | 0.462 | 0.527 | 0.502 | 0.474 | 0.453 | 0.543 | 0.49 | 0.585 | 0.516 | 0.671 | 0.561 | 0.908 | 0.724 | 1.166 | 0.823 | 1.15 | 0.82 | 1.102 | 0.841 | 0.632 | 0.596 | 1.324 | 0.858 |
| | Avg | 0.42875 | 0.42475 | 0.400 | 0.406 | 0.429 | 0.425 | 0.435 | 0.437 | 0.403 | 0.407 | 0.448 | 0.452 | 0.481 | 0.456 | 0.588 | 0.517 | 0.691 | 0.607 | 0.961 | 0.734 | 0.93 | 0.725 | 0.799 | 0.671 | 0.533 | 0.532 | 1.142 | 0.782 |
| ETTm2 | 96 | 0.187 | 0.269 | 0.187 | 0.267 | 0.189 | 0.28 | 0.209 | 0.308 | 0.193 | 0.292 | 0.203 | 0.287 | 0.192 | 0.274 | 0.255 | 0.339 | 0.435 | 0.507 | 0.365 | 0.453 | 0.77 | 0.642 | 0.658 | 0.619 | 0.243 | 0.342 | 2.041 | 1.073 |
| | 192 | 0.256 | 0.312 | 0.249 | 0.309 | 0.253 | 0.319 | 0.311 | 0.382 | 0.284 | 0.362 | 0.269 | 0.328 | 0.28 | 0.339 | 0.281 | 0.34 | 0.73 | 0.673 | 0.533 | 0.563 | 0.99 | 0.757 | 1.078 | 0.827 | 0.392 | 0.448 | 2.249 | 1.112 |
| | 336 | 0.314 | 0.347 | 0.321 | 0.351 | 0.314 | 0.357 | 0.442 | 0.466 | 0.369 | 0.427 | 0.325 | 0.366 | 0.334 | 0.361 | 0.339 | 0.372 | 1.201 | 0.845 | 1.363 | 0.887 | 1.33 | 0.872 | 1.549 | 0.972 | 0.932 | 0.724 | 2.568 | 1.238 |
| | 720 | 0.424 | 0.424 | 0.410 | 0.408 | 0.403 | 0.414 | 0.413 | 0.675 | 0.587 | 0.554 | 0.421 | 0.415 | 0.433 | 0.432 | 3.625 | 1.451 | 3.379 | 1.338 | 3.05 | 1.328 | 2.631 | 1.242 | 2.72 | 1.287 | | | | |
| | Avg | 0.295 | 0.335 | 0.291 | 0.333 | 0.293 | 0.342 | 0.409 | 0.436 | 0.35 | 0.401 | 0.305 | 0.349 | 0.306 | 0.347 | 0.327 | 0.371 | 1.498 | 0.869 | 1.41 | 0.81 | 1.54 | 0.9 | 1.479 | 0.915 | 0.735 | 0.598 | 2.395 | 1.177 |
| ETTh1 | 96 | 0.442 | 0.441 | 0.384 | 0.402 | 0.494 | 0.479 | 0.424 | 0.432 | 0.386 | 0.400 | 0.376 | 0.419 | 0.513 | 0.491 | 0.449 | 0.459 | 0.664 | 0.612 | 0.865 | 0.713 | 0.88 | 0.74 | 0.837 | 0.728 | 0.548 | 0.528 | 1.044 | 0.773 |
| | 192 | 0.490 | 0.474 | 0.436 | 0.429 | 0.538 | 0.504 | 0.475 | 0.462 | 0.437 | 0.432 | 0.420 | 0.448 | 0.534 | 0.504 | 0.5 | 0.482 | 0.79 | 0.681 | 1.008 | 0.792 | 1.04 | 0.824 | 0.923 | 0.766 | 0.542 | 0.526 | 1.217 | 0.832 |
| | 336 | 0.527 | 0.498 | 0.491 | 0.469 | 0.574 | 0.521 | 0.518 | 0.488 | 0.481 | 0.459 | 0.459 | 0.465 | 0.588 | 0.535 | 0.521 | 0.496 | 0.891 | 0.738 | 1.107 | 0.809 | 1.24 | 0.932 | 1.097 | 0.835 | 1.298 | 0.942 | 1.259 | 0.841 |
| | 720 | 0.530 | 0.509 | 0.521 | 0.500 | 0.562 | 0.535 | 0.547 | 0.533 | 0.519 | 0.516 | 0.506 | 0.507 | 0.643 | 0.616 | 0.514 | 0.512 | 0.963 | 0.782 | 1.181 | 0.865 | 1.14 | 0.852 | 1.257 | 0.889 | 0.721 | 0.659 | 1.271 | 0.838 |
| | Avg | 0.49725 | 0.4805 | 0.458 | 0.450 | 0.542 | 0.51 | 0.491 | 0.479 | 0.456 | 0.452 | 0.440 | 0.46 | 0.57 | 0.537 | 0.496 | 0.487 | 0.827 | 0.703 | 1.04 | 0.795 | 1.07 | 0.837 | 1.029 | 0.805 | 0.777 | 0.664 | 1.198 | 0.821 |
| ETTh2 | 96 | 0.339 | 0.376 | 0.34 | 0.374 | 0.34 | 0.391 | 0.397 | 0.437 | 0.333 | 0.387 | 0.358 | 0.397 | 0.476 | 0.458 | 0.346 | 0.388 | 0.645 | 0.597 | 3.755 | 1.525 | 2.12 | 1.197 | 2.626 | 1.317 | 1.616 | 1.036 | 2.522 | 1.278 |
| | 192 | 0.444 | 0.431 | 0.402 | 0.414 | 0.43 | 0.439 | 0.52 | 0.504 | 0.477 | 0.476 | 0.429 | 0.439 | 0.512 | 0.493 | 0.456 | 0.452 | 0.788 | 0.683 | 5.602 | 1.931 | 4.32 | 1.635 | 11.12 | 2.979 | 2.083 | 1.197 | 3.312 | 1.384 |
| | 336 | 0.475 | 0.457 | 0.452 | 0.452 | 0.485 | 0.479 | 0.626 | 0.559 | 0.594 | 0.541 | 0.496 | 0.487 | 0.552 | 0.551 | 0.482 | 0.486 | 0.907 | 0.747 | 4.721 | 1.835 | 1.12 | 1.604 | 9.323 | 2.769 | 1.439 | | 3.291 | 1.388 |
| | 720 | 0.458 | 0.460 | 0.462 | 0.468 | 0.5 | 0.497 | 0.863 | 0.672 | 0.831 | 0.657 | 0.463 | 0.474 | 0.562 | 0.56 | 0.515 | 0.511 | 0.963 | 0.783 | 3.647 | 1.625 | 3.19 | 1.54 | 3.874 | 1.697 | 2.576 | 1.363 | 3.257 | 1.357 |
| | Avg | 0.429 | 0.431 | 0.414 | 0.427 | 0.439 | 0.452 | 0.602 | 0.543 | 0.559 | 0.515 | 0.437 | 0.449 | 0.526 | 0.516 | 0.45 | 0.459 | 0.826 | 0.703 | 4.431 | 1.729 | 2.69 | 1.494 | 6.736 | 2.191 | 2.311 | 1.259 | 3.095 | 1.352 |
| Eelctricity | 96 | 0.170 | 0.273 | 0.168 | 0.272 | 0.187 | 0.304 | 0.207 | 0.307 | 0.197 | 0.282 | 0.193 | 0.308 | 0.169 | 0.273 | 0.201 | 0.317 | 0.386 | 0.449 | 0.274 | 0.368 | 0.29 | 0.357 | 0.312 | 0.402 | 0.3 | 0.392 | 0.375 | 0.437 |
| | 192 | 0.184 | 0.286 | 0.184 | 0.289 | 0.199 | 0.315 | 0.213 | 0.316 | 0.196 | 0.285 | 0.201 | 0.315 | 0.182 | 0.286 | 0.222 | 0.334 | 0.378 | 0.443 | 0.296 | 0.386 | 0.27 | 0.368 | 0.348 | 0.433 | 0.297 | 0.39 | 0.442 | 0.473 |
| | 336 | 0.196 | 0.299 | 0.198 | 0.300 | 0.212 | 0.329 | 0.23 | 0.333 | 0.209 | 0.301 | 0.214 | 0.329 | 0.2 | 0.304 | 0.231 | 0.338 | 0.376 | 0.443 | 0.3 | 0.394 | 0.28 | 0.38 | 0.35 | 0.433 | 0.317 | 0.403 | 0.439 | 0.473 |
| | 720 | 0.224 | 0.320 | 0.220 | 0.320 | 0.233 | 0.345 | 0.265 | 0.36 | 0.245 | 0.333 | 0.246 | 0.355 | 0.222 | 0.321 | 0.254 | 0.361 | 0.376 | 0.445 | 0.373 | 0.439 | 0.42 | 0.376 | 0.34 | 0.42 | 0.338 | 0.417 | 0.98 | 0.814 |
| | Avg | 0.194 | 0.295 | 0.192 | 0.295 | 0.208 | 0.323 | 0.229 | 0.329 | 0.212 | 0.3 | 0.214 | 0.327 | 0.193 | 0.296 | 0.227 | 0.338 | 0.379 | 0.445 | 0.311 | 0.397 | 0.27 | 0.37 | 0.338 | 0.422 | 0.313 | 0.401 | 0.559 | 0.549 |
| Traffic | 96 | 0.599 | 0.329 | 0.593 | 0.321 | 0.607 | 0.392 | 0.615 | 0.391 | 0.65 | 0.396 | 0.587 | 0.366 | 0.612 | 0.338 | 0.613 | 0.388 | 0.867 | 0.468 | 0.719 | 0.391 | 0.68 | 0.384 | 0.732 | 0.423 | 0.798 | 0.436 | 0.843 | 0.453 |
| | 192 | 0.621 | 0.338 | 0.617 | 0.336 | 0.621 | 0.399 | 0.601 | 0.382 | 0.598 | 0.37 | 0.604 | 0.373 | 0.613 | 0.340 | 0.616 | 0.382 | 0.869 | 0.467 | 0.696 | 0.379 | 0.69 | 0.39 | 0.733 | 0.42 | 0.849 | 0.481 | 0.847 | 0.453 |
| | 336 | 0.638 | 0.340 | 0.629 | 0.336 | 0.622 | 0.396 | 0.613 | 0.386 | 0.605 | 0.373 | 0.621 | 0.383 | 0.618 | 0.328 | 0.622 | 0.337 | 0.881 | 0.469 | 0.777 | 0.42 | 0.73 | 0.408 | 0.742 | 0.42 | 0.828 | 0.476 | 0.853 | 0.455 |
| | 720 | 0.648 | 0.356 | 0.64 | 0.350 | 0.632 | 0.396 | 0.658 | 0.407 | 0.645 | 0.394 | 0.626 | 0.382 | 0.653 | 0.355 | 0.66 | 0.408 | 0.896 | 0.473 | 0.864 | 0.472 | 0.72 | 0.396 | 0.755 | 0.423 | 0.854 | 0.489 | 1.5 | 0.805 |
| | Avg | 0.627 | 0.341 | 0.620 | 0.336 | 0.621 | 0.396 | 0.622 | 0.392 | 0.625 | 0.383 | 0.610 | 0.376 | 0.624 | 0.340 | 0.628 | 0.379 | 0.878 | 0.469 | 0.764 | 0.416 | 0.71 | 0.395 | 0.741 | 0.422 | 0.832 | 0.471 | 1.011 | 0.541 |
| Weather | 96 | 0.175 | 0.224 | 0.172 | 0.220 | 0.197 | 0.281 | 0.182 | 0.242 | 0.196 | 0.255 | 0.217 | 0.296 | 0.173 | 0.223 | 0.266 | 0.336 | 0.622 | 0.556 | 0.3 | 0.384 | 0.46 | 0.49 | 0.689 | 0.596 | 0.174 | 0.252 | 0.369 | 0.406 |
| | 192 | 0.226 | 0.266 | 0.219 | 0.261 | 0.237 | 0.312 | 0.227 | 0.287 | 0.237 | 0.296 | 0.276 | 0.336 | 0.245 | 0.285 | 0.307 | 0.367 | 0.739 | 0.624 | 0.598 | 0.544 | 0.66 | 0.589 | 0.752 | 0.638 | 0.238 | 0.313 | 0.416 | 0.435 |
| | 336 | 0.279 | 0.303 | 0.280 | 0.306 | 0.298 | 0.353 | 0.282 | 0.334 | 0.283 | 0.335 | 0.339 | 0.38 | 0.321 | 0.338 | 0.359 | 0.395 | 1.004 | 0.753 | 0.578 | 0.523 | 0.8 | 0.652 | 0.639 | 0.596 | 0.287 | 0.355 | 0.455 | 0.454 |
| | 720 | 0.358 | 0.355 | 0.365 | 0.359 | 0.352 | 0.288 | 0.352 | 0.386 | 0.345 | 0.381 | 0.403 | 0.428 | 0.414 | 0.41 | 0.419 | 0.428 | 1.42 | 0.934 | 1.059 | 0.741 | 0.87 | 0.675 | 1.13 | 0.792 | 0.384 | 0.415 | 0.535 | 0.52 |
| | Avg | 0.260 | 0.287 | 0.259 | 0.287 | 0.271 | 0.334 | 0.261 | 0.312 | 0.265 | 0.317 | 0.309 | 0.36 | 0.288 | 0.314 | 0.338 | 0.382 | 0.946 | 0.717 | 0.634 | 0.548 | 0.7 | 0.602 | 0.803 | 0.656 | 0.271 | 0.334 | 0.444 | 0.454 |
| Exchange | 96 | 0.130 | 0.264 | 0.107 | 0.234 | 0.085 | 0.204 | 0.116 | 0.262 | 0.088 | 0.218 | 0.148 | 0.278 | 0.111 | 0.237 | 0.197 | 0.323 | 1.748 | 1.105 | 0.847 | 0.752 | 0.97 | 0.812 | 1.065 | 0.829 | 0.395 | 0.474 | 1.453 | 1.049 |
| | 192 | 0.181 | 0.314 | 0.226 | 0.344 | 0.182 | 0.303 | 0.215 | 0.359 | 0.176 | 0.315 | 0.271 | 0.38 | 0.219 | 0.335 | 0.3 | 0.369 | 1.874 | 1.151 | 1.204 | 0.895 | 1.04 | 0.851 | 1.188 | 0.906 | 0.776 | 0.698 | 1.846 | 1.179 |
| | 336 | 0.374 | 0.442 | 0.367 | 0.448 | 0.348 | 0.428 | 0.377 | 0.466 | 0.313 | 0.427 | 0.46 | 0.5 | 0.421 | 0.476 | 0.509 | 0.524 | 1.943 | 1.172 | 1.672 | 1.036 | 1.66 | 1.081 | 1.357 | 0.976 | 1.029 | 0.797 | 2.136 | 1.231 |
| | 720 | 0.963 | 0.749 | 0.964 | 0.746 | 1.025 | 0.774 | 0.831 | 0.699 | 0.839 | 0.695 | 1.195 | 0.841 | 0.769 | 1.447 | 0.941 | 1.285 | 1.206 | 2.478 | 1.31 | 1.94 | 1.127 | 1.51 | 1.016 | 2.283 | 1.222 | 2.984 | 1.427 | |
| | Avg | 0.412 | 0.44225 | 0.416 | 0.443 | 0.41 | 0.427 | 0.385 | 0.447 | 0.354 | 0.414 | 0.519 | 0.5 | 0.461 | 0.454 | 0.613 | 0.539 | 1.913 | 1.159 | 1.55 | 0.998 | 1.4 | 0.968 | 1.28 | 0.932 | 1.121 | 0.798 | 2.105 | 1.221 |
| ILL | 24 | 2.625 | 0.957 | 2.317 | 0.934 | 2.527 | 1.02 | 8.313 | 2.144 | 2.398 | 1.04 | 3.228 | 1.26 | 2.294 | 0.945 | 3.483 | 1.287 | 7.394 | 2.012 | 5.764 | 1.677 | 4.48 | 1.444 | 4.4 | 1.382 | 4.381 | 1.425 | 5.914 | 1.734 |
| | 36 | 2.768 | 1.015 | 1.972 | 0.920 | 2.615 | 1.007 | 6.631 | 1.902 | 2.646 | 1.088 | 2.679 | 1.08 | 1.825 | 0.848 | 3.103 | 1.148 | 7.551 | 2.031 | 4.755 | 1.467 | 4.8 | 1.467 | 4.783 | 1.448 | 4.442 | 1.416 | 6.631 | 1.845 |
| | 48 | 2.234 | 0.937 | 2.238 | 0.94 | 2.359 | 0.972 | 7.299 | 1.982 | 2.614 | 1.086 | 2.622 | 1.078 | 2.01 | 0.900 | 2.669 | 1.085 | 7.662 | 2.057 | 4.763 | 1.469 | 4.8 | 1.468 | 4.832 | 1.465 | 4.559 | 1.443 | 6.736 | 1.857 |
| | 60 | 2.205 | 0.971 | 2.027 | 0.928 | 2.487 | 1.016 | 7.283 | 1.985 | 2.804 | 1.146 | 2.857 | 1.157 | 2.178 | 0.963 | 2.77 | 1.125 | 7.931 | 2.1 | 5.264 | 1.564 | 5.28 | 1.56 | 4.882 | 1.483 | 4.651 | 1.474 | 6.87 | 1.879 |
| | Avg | 2.458 | 0.97 | 2.139 | 0.931 | 2.497 | 1.004 | 7.382 | 2.003 | 2.616 | 1.09 | 2.847 | 1.144 | 2.077 | 0.914 | 3.006 | 1.161 | 7.635 | 2.05 | 5.137 | 1.544 | 4.84 | 1.485 | 4.724 | 1.445 | 4.508 | 1.44 | 6.538 | 1.829 |
| Count 1st | | 11 1st | | 30 1st | | 4 1st | | 1 1st | | 4 1st | | 13 1st | | 6 1st | | 7 1st | | | | | | | | | | | | | |

Table 9: The result of short-term forecasting task( order-preserving).

| Models | SpecAR-Net (ours) | TimesNet (2023) | N-HiTS (2022) | N-BEATS (2019) | ETS. (2022) | LightTS (2022) | Dlinear (2023) | FED. (2022) | Stationary (2022a) | Auto. (2021) | Pyra. (2021a) | In. (2021) | LogTrans (2019) | Re. (2020) | LSTM (1997) | TCN (2019) | LSSL (2022) |
|---|---|---|---|---|---|---|---|---|---|---|---|---|---|---|---|---|---|
| **Yearly** SMAPE | **13.27** | **13.387** | 13.418 | 13.436 | 18.009 | 14.247 | 16.965 | 13.728 | 13.717 | 13.974 | 15.53 | 14.727 | 17.107 | 16.169 | 176.04 | 14.92 | 61.675 |
| MASE | **2.983** | **2.996** | 3.045 | 3.043 | 4.487 | 3.109 | 4.283 | 3.048 | 3.078 | 3.134 | 3.711 | 3.418 | 4.177 | 3.8 | 31.033 | 3.364 | 19.953 |
| OWA | **0.781** | **0.786** | 0.793 | 0.794 | 1.115 | 0.827 | 1.058 | 0.803 | 0.807 | 0.822 | 0.942 | 0.881 | 1.049 | 0.973 | 9.29 | 0.88 | 4.397 |
| **Quarterly** SMAPE | **10.071** | **10.100** | 10.202 | 10.124 | 13.376 | 11.364 | 12.145 | 10.792 | 10.958 | 11.338 | 15.449 | 11.36 | 13.207 | 13.313 | 172.808 | 11.122 | 65.999 |
| MASE | **1.174** | 1.182 | 1.194 | **1.169** | 1.906 | 1.328 | 1.52 | 1.283 | 1.325 | 1.365 | 2.35 | 1.401 | 1.827 | 1.775 | 19.753 | 1.36 | 17.662 |
| OWA | **0.885** | 0.89 | 0.899 | **0.886** | 1.302 | 1 | 1.106 | 0.958 | 0.981 | 1.012 | 1.558 | 1.027 | 1.266 | 1.252 | 15.049 | 1.001 | 9.436 |
| **Monthly** SMAPE | 12.784 | **12.67** | 12.791 | **12.677** | 14.588 | 14.014 | 13.514 | 14.26 | 13.917 | 13.958 | 17.642 | 14.062 | 16.149 | 20.128 | 143.237 | 15.626 | 64.664 |
| MASE | 0.944 | **0.933** | 0.969 | **0.937** | 1.368 | 1.053 | 1.037 | 1.102 | 1.097 | 1.103 | 1.913 | 1.141 | 1.66 | 2.614 | 16.551 | 1.274 | 16.245 |
| OWA | 0.887 | **0.878** | 0.899 | **0.880** | 1.149 | 0.981 | 0.956 | 1.012 | 0.998 | 1.002 | 1.511 | 1.024 | 1.34 | 1.927 | 12.747 | 1.141 | 9.879 |
| **Others** SMAPE | **4.762** | **4.891** | 5.061 | 4.925 | 7.267 | 15.88 | 6.709 | 4.954 | 6.302 | 5.485 | 24.786 | 24.46 | 23.236 | 32.491 | 186.282 | 7.186 | 121.844 |
| MASE | **3.212** | 3.302 | **3.216** | 3.391 | 5.24 | 11.434 | 4.953 | 3.264 | 4.064 | 3.865 | 18.581 | 20.96 | 16.288 | 33.355 | 119.294 | 4.677 | 91.65 |
| OWA | **1.008** | **1.035** | 1.04 | 1.053 | 1.591 | 3.474 | 1.487 | 1.036 | 1.304 | 1.187 | 5.538 | 5.879 | 5.013 | 8.679 | 38.411 | 1.494 | 27.273 |
| **W-Average** SMAPE | **11.844** | **11.829** | 11.927 | 11.851 | 14.718 | 13.525 | 13.639 | 12.84 | 12.78 | 12.909 | 16.987 | 14.086 | 16.018 | 18.2 | 160.031 | 13.961 | 67.156 |
| MASE | **1.582** | **1.585** | 1.613 | 1.599 | 2.408 | 2.111 | 2.095 | 1.701 | 1.756 | 1.771 | 3.265 | 2.718 | 3.01 | 4.223 | 25.788 | 1.945 | 21.208 |
| OWA | **0.85** | **0.851** | 0.861 | 0.855 | 1.172 | 1.051 | 1.051 | 0.918 | 0.93 | 0.939 | 1.48 | 1.23 | 1.378 | 1.775 | 12.642 | 1.023 | 8.021 |

Table 10: The result of long-term forecasting task(order-preserving).

| Models | SpecAR-Net (ours) | | TimesNet (2023) | | ETSformer (2022) | | LightTS (2022) | | Dlinear (2023) | | FEDformer (2022) | | Stationary (2022a) | | Autoformer (2021) | | Pyraformer (2021a) | | Informer (2021) | | LogTrans (2019) | | Reformer (2020) | | LSSL (2022) | | LSTM (1997) | |
|---|---|---|---|---|---|---|---|---|---|---|---|---|---|---|---|---|---|---|---|---|---|---|---|---|---|---|---|---|
| Metrics | MSE | MAE | MSE | MAE | MSE | MAE | MSE | MAE | MSE | MAE | MSE | MAE | MSE | MAE | MSE | MAE | MSE | MAE | MSE | MAE | MSE | MAE | MSE | MAE | MSE | MAE | MSE | MAE |
| **ETTm1** 96 | **0.323** | **0.365** | **0.338** | 0.375 | 0.375 | 0.398 | 0.374 | 0.4 | 0.345 | **0.372** | 0.379 | 0.419 | 0.386 | 0.398 | 0.505 | 0.475 | 0.543 | 0.51 | 0.672 | 0.571 | 0.6 | 0.546 | 0.538 | 0.528 | 0.45 | 0.477 | 0.863 | 0.664 |
| 192 | **0.375** | 0.395 | **0.374** | **0.387** | 0.408 | 0.41 | 0.4 | 0.407 | 0.38 | **0.389** | 0.426 | 0.441 | 0.459 | 0.444 | 0.553 | 0.496 | 0.557 | 0.537 | 0.795 | 0.669 | 0.837 | 0.7 | 0.658 | 0.592 | 0.469 | 0.481 | 1.113 | 0.776 |
| 336 | **0.413** | 0.417 | 0.41 | **0.411** | 0.435 | 0.428 | 0.438 | 0.438 | **0.413** | **0.413** | 0.445 | 0.459 | 0.495 | 0.464 | 0.621 | 0.537 | 0.754 | 0.655 | 1.212 | 0.871 | 1.124 | 0.832 | 0.898 | 0.721 | 0.583 | 0.574 | 1.267 | 0.832 |
| 720 | 0.482 | 0.451 | 0.478 | **0.450** | 0.499 | 0.462 | 0.527 | 0.502 | 0.474 | 0.453 | 0.543 | 0.49 | 0.585 | 0.516 | 0.671 | 0.561 | 0.908 | 0.724 | 1.166 | 0.823 | 1.153 | 0.82 | 1.102 | 0.841 | 0.632 | 0.596 | 1.324 | 0.858 |
| Avg | **0.398** | **0.407** | 0.400 | 0.406 | 0.429 | 0.425 | 0.435 | 0.437 | 0.403 | 0.407 | 0.448 | 0.452 | 0.481 | 0.456 | 0.588 | 0.517 | 0.691 | 0.607 | 0.961 | 0.734 | 0.929 | 0.725 | 0.799 | 0.671 | 0.533 | 0.532 | 1.142 | 0.782 |
| **ETTm2** 96 | **0.183** | **0.265** | 0.187 | 0.267 | 0.189 | 0.28 | 0.209 | 0.308 | 0.193 | 0.292 | 0.203 | 0.287 | 0.192 | 0.274 | 0.255 | 0.339 | 0.435 | 0.507 | 0.365 | 0.453 | 0.768 | 0.642 | 0.658 | 0.619 | 0.243 | 0.342 | 2.041 | 1.073 |
| 192 | **0.251** | **0.308** | 0.249 | 0.309 | 0.253 | 0.319 | 0.311 | 0.382 | 0.284 | 0.362 | 0.269 | 0.328 | 0.28 | 0.339 | 0.281 | 0.34 | 0.73 | 0.673 | 0.533 | 0.563 | 0.989 | 0.757 | 1.078 | 0.827 | 0.392 | 0.448 | 2.249 | 1.112 |
| 336 | **0.311** | **0.348** | 0.321 | 0.351 | 0.314 | 0.357 | 0.442 | 0.466 | 0.369 | 0.427 | 0.325 | 0.366 | 0.334 | 0.361 | 0.339 | 0.372 | 1.201 | 0.845 | 1.363 | 0.887 | 1.334 | 0.872 | 1.549 | 0.972 | 0.932 | 0.724 | 2.568 | 1.238 |
| 720 | 0.417 | **0.408** | 0.408 | **0.403** | 0.414 | 0.413 | 0.675 | 0.587 | 0.554 | 0.522 | 0.421 | 0.415 | 0.417 | 0.413 | 0.433 | 0.432 | 3.625 | 1.451 | 3.379 | 1.338 | 3.048 | 1.328 | 2.631 | 1.242 | 1.372 | 0.879 | 2.72 | 1.287 |
| Avg | **0.291** | **0.332** | 0.291 | 0.333 | 0.293 | 0.342 | 0.409 | 0.436 | 0.35 | 0.401 | 0.305 | 0.349 | 0.306 | 0.347 | 0.327 | 0.371 | 1.498 | 0.869 | 1.41 | 0.81 | 1.535 | 0.9 | 1.479 | 0.915 | 0.735 | 0.598 | 2.395 | 1.177 |
| **ETTh1** 96 | 0.399 | 0.416 | **0.384** | **0.402** | 0.494 | 0.479 | 0.424 | 0.432 | 0.386 | 0.4 | **0.376** | 0.419 | 0.513 | 0.491 | 0.449 | 0.459 | 0.664 | 0.612 | 0.865 | 0.713 | 0.878 | 0.74 | 0.837 | 0.728 | 0.548 | 0.528 | 1.044 | 0.773 |
| 192 | 0.448 | 0.446 | **0.436** | **0.429** | 0.538 | 0.504 | 0.475 | 0.462 | 0.437 | **0.432** | 0.42 | 0.448 | 0.534 | 0.504 | 0.5 | 0.482 | 0.79 | 0.681 | 1.008 | 0.792 | 1.037 | 0.824 | 0.923 | 0.766 | 0.542 | 0.526 | 1.217 | 0.832 |
| 336 | 0.482 | **0.464** | 0.491 | 0.469 | 0.574 | 0.521 | 0.518 | 0.488 | **0.481** | **0.459** | 0.459 | 0.465 | 0.588 | 0.535 | 0.521 | 0.496 | 0.891 | 0.738 | 1.107 | 0.809 | 1.238 | 0.932 | 1.097 | 0.835 | 1.298 | 0.942 | 1.259 | 0.841 |
| 720 | **0.504** | **0.494** | 0.521 | **0.500** | 0.562 | 0.535 | 0.547 | 0.533 | 0.519 | 0.516 | **0.506** | 0.507 | 0.643 | 0.616 | 0.514 | 0.512 | 0.963 | 0.782 | 1.181 | 0.865 | 1.135 | 0.852 | 1.257 | 0.889 | 0.721 | 0.659 | 1.271 | 0.838 |
| Avg | 0.458 | 0.455 | 0.458 | 0.450 | 0.542 | 0.51 | 0.491 | 0.479 | **0.456** | **0.452** | **0.440** | 0.46 | 0.57 | 0.537 | 0.496 | 0.487 | 0.827 | 0.703 | 1.04 | 0.795 | 1.072 | 0.837 | 1.029 | 0.805 | 0.777 | 0.664 | 1.198 | 0.821 |
| **ETTh2** 96 | 0.352 | 0.385 | **0.340** | **0.374** | 0.34 | 0.391 | 0.397 | 0.437 | **0.333** | **0.387** | 0.358 | 0.397 | 0.476 | 0.458 | 0.346 | 0.388 | 0.645 | 0.597 | 3.755 | 1.525 | 2.116 | 1.197 | 2.626 | 1.317 | 1.616 | 1.036 | 2.522 | 1.278 |
| 192 | **0.417** | **0.420** | 0.402 | 0.414 | 0.43 | 0.439 | 0.52 | 0.504 | 0.477 | 0.476 | 0.429 | 0.439 | 0.512 | 0.493 | 0.456 | 0.452 | 0.788 | 0.683 | 5.602 | 1.931 | 4.315 | 1.635 | 11.12 | 2.979 | 2.083 | 1.197 | 3.312 | 1.384 |
| 336 | **0.447** | **0.447** | 0.452 | 0.452 | 0.485 | 0.479 | 0.626 | 0.559 | 0.594 | 0.541 | 0.496 | 0.487 | 0.552 | 0.551 | 0.482 | 0.486 | 0.907 | 0.747 | 4.721 | 1.835 | 1.124 | 1.604 | 9.323 | 2.769 | 2.97 | 1.439 | 3.291 | 1.388 |
| 720 | **0.449** | **0.456** | 0.462 | 0.468 | 0.5 | 0.497 | 0.863 | 0.672 | 0.831 | 0.657 | 0.463 | 0.474 | 0.562 | 0.56 | 0.515 | 0.511 | 0.963 | 0.783 | 3.647 | 1.625 | 3.188 | 1.54 | 3.874 | 1.697 | 2.576 | 1.363 | 3.257 | 1.357 |
| Avg | **0.416** | **0.427** | 0.414 | 0.427 | 0.439 | 0.452 | 0.602 | 0.543 | 0.559 | 0.515 | 0.437 | **0.449** | 0.526 | 0.516 | 0.45 | 0.459 | 0.826 | 0.703 | 4.431 | 1.729 | 2.686 | 1.494 | 6.736 | 2.191 | 2.311 | 1.259 | 3.095 | 1.352 |
| **Eelctricity** 96 | **0.165** | **0.27** | **0.168** | 0.272 | 0.187 | 0.304 | 0.207 | 0.307 | 0.197 | 0.282 | 0.193 | 0.308 | 0.169 | 0.273 | 0.201 | 0.317 | 0.386 | 0.449 | 0.274 | 0.368 | 0.258 | 0.357 | 0.312 | 0.402 | 0.3 | 0.392 | 0.375 | 0.437 |
| 192 | 0.186 | 0.287 | **0.184** | 0.289 | 0.199 | 0.315 | 0.213 | 0.316 | 0.196 | **0.285** | 0.201 | 0.315 | **0.182** | 0.286 | 0.222 | 0.334 | 0.378 | 0.443 | 0.296 | 0.386 | 0.266 | 0.368 | 0.348 | 0.433 | 0.297 | 0.39 | 0.442 | 0.473 |
| 336 | **0.195** | **0.298** | 0.198 | 0.3 | 0.212 | 0.329 | 0.23 | 0.333 | 0.209 | 0.301 | 0.214 | 0.329 | 0.2 | 0.304 | 0.231 | 0.338 | 0.376 | 0.443 | 0.3 | 0.394 | 0.28 | 0.38 | 0.35 | 0.433 | 0.317 | 0.403 | 0.439 | 0.473 |
| 720 | 0.223 | 0.321 | 0.220 | 0.32 | 0.233 | 0.345 | 0.265 | 0.36 | 0.245 | 0.333 | 0.246 | 0.355 | **0.222** | 0.321 | 0.254 | 0.361 | 0.376 | 0.445 | 0.373 | 0.439 | 0.283 | 0.376 | 0.34 | 0.42 | 0.338 | 0.417 | 0.98 | 0.814 |
| Avg | **0.192** | 0.294 | **0.192** | 0.295 | 0.208 | 0.323 | 0.229 | 0.329 | 0.212 | 0.3 | 0.214 | 0.327 | **0.193** | 0.296 | 0.227 | 0.338 | 0.379 | 0.445 | 0.311 | 0.397 | 0.272 | 0.37 | 0.338 | 0.422 | 0.313 | 0.401 | 0.559 | 0.549 |
| **Traffic** 96 | 0.601 | **0.323** | **0.593** | **0.321** | 0.607 | 0.392 | 0.615 | 0.391 | 0.65 | 0.396 | **0.587** | 0.366 | 0.612 | 0.338 | 0.613 | 0.388 | 0.867 | 0.468 | 0.719 | 0.391 | 0.684 | 0.384 | 0.732 | 0.423 | 0.798 | 0.436 | 0.843 | 0.453 |
| 192 | 0.616 | **0.329** | 0.617 | **0.336** | 0.621 | 0.399 | 0.601 | 0.382 | **0.598** | 0.37 | 0.604 | 0.373 | 0.613 | **0.340** | 0.616 | 0.382 | 0.869 | 0.467 | 0.696 | 0.379 | 0.685 | 0.39 | 0.733 | 0.42 | 0.849 | 0.481 | 0.847 | 0.453 |
| 336 | 0.633 | 0.337 | 0.629 | 0.336 | 0.622 | 0.396 | 0.613 | 0.386 | **0.605** | 0.373 | 0.621 | 0.383 | 0.618 | **0.328** | 0.622 | 0.337 | 0.881 | 0.469 | 0.777 | 0.42 | 0.734 | 0.408 | 0.742 | 0.42 | 0.828 | 0.476 | 0.853 | 0.455 |
| 720 | 0.651 | 0.352 | 0.64 | 0.35 | **0.632** | 0.396 | 0.658 | 0.407 | 0.645 | 0.394 | **0.626** | 0.382 | 0.653 | **0.355** | 0.66 | 0.408 | 0.896 | 0.473 | 0.864 | 0.472 | 0.717 | 0.396 | 0.755 | 0.423 | 0.854 | 0.489 | 1.5 | 0.805 |
| Avg | 0.625 | **0.335** | **0.620** | 0.336 | 0.621 | 0.396 | 0.622 | 0.392 | 0.625 | 0.383 | **0.610** | 0.376 | 0.624 | 0.34 | 0.628 | 0.379 | 0.878 | 0.469 | 0.764 | 0.416 | 0.705 | 0.395 | 0.741 | 0.422 | 0.832 | 0.471 | 1.011 | 0.541 |
| **Weather** 96 | **0.173** | **0.222** | **0.172** | 0.22 | 0.197 | 0.281 | 0.182 | 0.242 | 0.196 | 0.255 | 0.217 | 0.296 | **0.173** | 0.223 | 0.266 | 0.336 | 0.622 | 0.556 | 0.3 | 0.384 | 0.458 | 0.49 | 0.689 | 0.596 | 0.174 | 0.252 | 0.369 | 0.406 |
| 192 | **0.220** | 0.261 | 0.219 | 0.261 | 0.237 | 0.312 | 0.227 | **0.287** | 0.22 | 0.292 | 0.276 | 0.336 | 0.245 | 0.307 | 0.307 | 0.367 | 0.739 | 0.624 | 0.598 | 0.544 | 0.658 | 0.589 | 0.752 | 0.638 | 0.238 | 0.313 | 0.416 | 0.435 |
| 336 | **0.277** | **0.302** | 0.28 | 0.306 | 0.298 | 0.353 | 0.282 | 0.334 | 0.283 | 0.335 | 0.339 | 0.38 | 0.321 | 0.338 | 0.359 | 0.395 | 1.004 | 0.753 | 0.578 | 0.523 | 0.797 | 0.652 | 0.639 | 0.596 | 0.287 | 0.355 | 0.455 | 0.454 |
| 720 | 0.357 | **0.352** | 0.365 | 0.359 | 0.352 | 0.288 | 0.352 | 0.386 | **0.345** | 0.381 | 0.403 | 0.428 | 0.414 | 0.41 | 0.419 | 0.428 | 1.42 | 0.934 | 1.059 | 0.741 | 0.869 | 0.675 | 1.13 | 0.792 | 0.384 | 0.415 | 0.535 | 0.52 |
| Avg | **0.257** | **0.284** | 0.259 | 0.287 | 0.271 | 0.334 | 0.261 | 0.312 | 0.265 | 0.317 | 0.309 | 0.36 | 0.288 | 0.314 | 0.338 | 0.382 | 0.946 | 0.717 | 0.634 | 0.548 | 0.696 | 0.602 | 0.803 | 0.656 | 0.271 | 0.334 | 0.444 | 0.454 |
| **Exchange** 96 | 0.109 | 0.238 | 0.107 | 0.234 | **0.085** | **0.204** | 0.116 | 0.262 | **0.088** | **0.218** | 0.148 | 0.278 | 0.111 | 0.237 | 0.197 | 0.323 | 1.748 | 1.105 | 0.847 | 0.752 | 0.968 | 0.812 | 1.065 | 0.829 | 0.395 | 0.474 | 1.453 | 1.049 |
| 192 | 0.198 | 0.323 | 0.226 | 0.344 | **0.182** | **0.303** | 0.215 | 0.359 | **0.176** | **0.315** | 0.271 | 0.38 | 0.219 | 0.335 | 0.3 | 0.369 | 1.874 | 1.151 | 1.204 | 0.895 | 1.04 | 0.851 | 1.188 | 0.906 | 0.776 | 0.698 | 1.846 | 1.179 |
| 336 | **0.337** | **0.424** | 0.367 | 0.448 | 0.348 | 0.428 | 0.377 | 0.466 | **0.313** | **0.427** | 0.46 | 0.5 | 0.421 | 0.476 | 0.509 | 0.524 | 1.943 | 1.172 | 1.672 | 1.036 | 1.659 | 1.081 | 1.357 | 0.976 | 1.029 | 0.797 | 2.136 | 1.231 |
| 720 | 0.89 | 0.716 | 0.964 | 0.746 | 1.025 | 0.774 | **0.831** | **0.699** | 0.839 | **0.695** | 1.195 | 0.841 | 0.769 | 1.447 | 2.085 | 1.206 | 2.478 | 1.31 | 1.941 | 1.127 | 1.51 | 1.016 | 2.283 | 1.222 | 2.984 | 1.427 | | |
| Avg | **0.384** | **0.425** | 0.416 | 0.443 | 0.41 | 0.427 | 0.385 | 0.447 | **0.354** | **0.414** | 0.519 | 0.5 | 0.461 | 0.613 | 0.539 | 1.913 | 1.159 | 1.55 | 0.998 | 1.402 | 0.968 | 1.28 | 0.932 | 1.121 | 0.798 | 2.105 | 1.221 | |
| **ILL** 24 | **1.859** | **0.858** | 2.317 | 0.934 | 2.527 | 1.02 | 8.313 | 2.144 | 2.398 | 1.04 | 3.228 | 1.26 | **2.294** | 0.945 | 3.483 | 1.287 | 7.394 | 2.012 | 5.764 | 1.677 | 4.48 | 1.444 | 4.4 | 1.382 | 4.381 | 1.425 | 5.914 | 1.734 |
| 36 | 2.454 | 0.985 | **1.972** | **0.920** | 2.615 | 1.007 | 6.631 | 1.902 | 2.646 | 1.088 | 2.679 | 1.08 | **1.825** | **0.848** | 3.103 | 1.148 | 7.551 | 2.031 | 4.755 | 1.467 | 4.799 | 1.467 | 4.783 | 1.448 | 4.442 | 1.416 | 6.631 | 1.845 |
| 48 | **1.919** | **0.846** | 2.238 | 0.94 | 2.359 | 0.972 | 7.299 | 1.982 | 2.614 | 1.086 | 2.622 | 1.078 | **2.010** | **0.900** | 2.669 | 1.085 | 7.662 | 2.057 | 4.763 | 1.469 | 4.8 | 1.468 | 4.832 | 1.465 | 4.559 | 1.443 | 6.736 | 1.857 |
| 60 | **1.972** | **0.921** | 2.027 | 0.928 | 2.487 | 1.016 | 7.283 | 1.985 | 2.804 | 1.146 | 2.857 | 1.157 | 2.178 | 0.963 | 2.77 | 1.125 | 7.931 | 2.1 | 5.264 | 1.564 | 5.278 | 1.56 | 4.882 | 1.483 | 4.651 | 1.474 | 6.87 | 1.879 |
| Avg | **2.051** | **0.903** | 2.139 | 0.931 | 2.497 | 1.004 | 7.382 | 2.003 | 2.616 | 1.09 | 2.847 | 1.144 | **2.077** | **0.914** | 3.006 | 1.161 | 7.635 | 2.05 | 5.137 | 1.544 | 4.839 | 1.485 | 4.724 | 1.445 | 4.508 | 1.44 | 6.538 | 1.829 |
| Count 1st | 29 | | 21 | | 3 | | 1 | | 11 | | 5 | | 4 | | | | | | | | | | | | | | | |

## Table 11: The result of imputation task.

| Models | SpecAR-Net (ours) | | TimesNet (2023) | | ETS. (2022) | | LightTS (2022) | | DLinear (2023) | | FED. (2022) | | Stationary (2022a) | | Auto. (2021) | | Pyra. (2021a) | | In. (2021) | | LogTrans (2019) | | Re. (2020) | | LSTM (1997) | | TCN (2019) | | LSSL (2022) | |
|---|---|---|---|---|---|---|---|---|---|---|---|---|---|---|---|---|---|---|---|---|---|---|---|---|---|---|---|---|---|---|---|
| MaskRate | MSE | MAE | MSE | MAE | MSE | MAE | MSE | MAE | MSE | MAE | MSE | MAE | MSE | MAE | MSE | MAE | MSE | MAE | MSE | MAE | MSE | MAE | MSE | MAE | MSE | MAE | MSE | MAE | MSE | MAE |
| **ETTm1** 12.50% | **0.019** | **0.093** | 0.019 | 0.092 | 0.067 | 0.188 | 0.075 | 0.18 | 0.058 | 0.162 | 0.035 | 0.135 | 0.026 | 0.107 | 0.034 | 0.124 | 0.67 | 0.541 | 0.047 | 0.155 | 0.041 | 0.141 | 0.032 | 0.126 | 0.974 | 0.78 | 0.51 | 0.493 | 0.101 | 0.231 |
| 25% | **0.024** | **0.104** | 0.023 | 0.101 | 0.096 | 0.229 | 0.093 | 0.206 | 0.08 | 0.193 | 0.052 | 0.166 | 0.032 | 0.119 | 0.046 | 0.144 | 0.689 | 0.553 | 0.063 | 0.18 | 0.044 | 0.144 | 0.042 | 0.146 | 1.032 | 0.807 | 0.518 | 0.5 | 0.106 | 0.235 |
| 37.50% | **0.029** | **0.113** | 0.029 | 0.111 | 0.133 | 0.271 | 0.113 | 0.231 | 0.103 | 0.219 | 0.069 | 0.191 | 0.039 | 0.131 | 0.057 | 0.161 | 0.737 | 0.581 | 0.079 | 0.2 | 0.052 | 0.158 | 0.063 | 0.182 | 0.999 | 0.792 | 0.516 | 0.499 | 0.116 | 0.246 |
| 50% | **0.036** | **0.126** | 0.036 | 0.124 | 0.186 | 0.323 | 0.134 | 0.255 | 0.132 | 0.248 | 0.089 | 0.218 | 0.047 | 0.145 | 0.067 | 0.174 | 0.77 | 0.605 | 0.093 | 0.218 | 0.063 | 0.173 | 0.082 | 0.208 | 0.952 | 0.763 | 0.519 | 0.496 | 0.129 | 0.26 |
| Avg | **0.027** | **0.109** | 0.027 | 0.107 | 0.12 | 0.253 | 0.104 | 0.218 | 0.093 | 0.206 | 0.062 | 0.177 | 0.036 | 0.126 | 0.051 | 0.15 | 0.717 | 0.57 | 0.071 | 0.188 | 0.05 | 0.154 | 0.055 | 0.166 | 0.989 | 0.786 | 0.516 | 0.497 | 0.113 | 0.254 |
| **ETTm2** 12.50% | **0.018** | **0.080** | 0.018 | 0.08 | 0.108 | 0.239 | 0.034 | 0.127 | 0.062 | 0.166 | 0.056 | 0.159 | 0.021 | 0.088 | 0.023 | 0.092 | 0.394 | 0.47 | 0.133 | 0.27 | 0.103 | 0.229 | 0.108 | 0.228 | 1.013 | 0.805 | 0.307 | 0.441 | 0.15 | 0.298 |
| 25% | **0.020** | **0.084** | 0.02 | 0.085 | 0.164 | 0.294 | 0.042 | 0.143 | 0.085 | 0.196 | 0.08 | 0.195 | 0.024 | 0.096 | 0.026 | 0.101 | 0.421 | 0.482 | 0.135 | 0.272 | 0.120 | 0.248 | 0.136 | 0.262 | 1.039 | 0.814 | 0.263 | 0.402 | 0.159 | 0.306 |
| 37.50% | **0.022** | **0.090** | 0.023 | 0.091 | 0.237 | 0.356 | 0.051 | 0.159 | 0.106 | 0.222 | 0.11 | 0.231 | 0.027 | 0.103 | 0.03 | 0.108 | 0.478 | 0.521 | 0.155 | 0.293 | 0.138 | 0.26 | 0.175 | 0.3 | 0.917 | 0.744 | 0.25 | 0.396 | 0.18 | 0.321 |
| 50% | **0.025** | **0.096** | 0.026 | 0.098 | 0.323 | 0.421 | 0.059 | 0.174 | 0.131 | 0.247 | 0.156 | 0.276 | 0.03 | 0.108 | 0.035 | 0.119 | 0.568 | 0.56 | 0.2 | 0.333 | 0.117 | 0.247 | 0.211 | 0.329 | 1.14 | 0.835 | 0.246 | 0.389 | 0.21 | 0.353 |
| **ETTh1** 12.50% | **0.055** | **0.163** | 0.057 | 0.159 | 0.126 | 0.263 | 0.24 | 0.345 | 0.151 | 0.267 | 0.07 | 0.19 | 0.06 | 0.165 | 0.074 | 0.182 | 0.857 | 0.609 | 0.114 | 0.234 | 0.229 | 0.33 | 0.074 | 0.194 | 1.265 | 0.896 | 0.599 | 0.554 | 0.422 | 0.461 |
| 25% | **0.064** | **0.174** | 0.069 | 0.178 | 0.169 | 0.304 | 0.265 | 0.364 | 0.18 | 0.292 | 0.106 | 0.236 | 0.08 | 0.189 | 0.09 | 0.203 | 0.829 | 0.672 | 0.14 | 0.262 | 0.207 | 0.323 | 0.102 | 0.227 | 1.262 | 0.883 | 0.61 | 0.567 | 0.412 | 0.456 |
| 37.50% | **0.083** | **0.195** | 0.084 | 0.196 | 0.22 | 0.347 | 0.296 | 0.382 | 0.215 | 0.318 | 0.124 | 0.258 | 0.102 | 0.212 | 0.109 | 0.222 | 0.83 | 0.675 | 0.174 | 0.293 | 0.21 | 0.328 | 0.135 | 0.261 | 1.2 | 0.867 | 0.628 | 0.577 | 0.421 | 0.461 |
| 50% | 0.106 | 0.217 | 0.102 | 0.215 | 0.293 | 0.402 | 0.334 | 0.404 | 0.257 | 0.347 | 0.165 | 0.299 | 0.133 | 0.24 | 0.137 | 0.248 | 0.854 | 0.691 | 0.215 | 0.325 | 0.23 | 0.348 | 0.179 | 0.298 | 1.174 | 0.849 | 0.648 | 0.587 | 0.43 | 0.473 |
| Avg | **0.077** | **0.187** | 0.078 | 0.187 | 0.202 | 0.329 | 0.284 | 0.373 | 0.201 | 0.306 | 0.117 | 0.246 | 0.094 | **0.201** | 0.103 | 0.214 | 0.842 | 0.682 | 0.161 | 0.279 | 0.219 | 0.332 | 0.122 | 0.245 | 1.225 | 0.873 | 0.621 | 0.571 | 0.424 | 0.481 |
| **ETTh2** 12.50% | **0.040** | **0.132** | 0.040 | 0.130 | 0.187 | 0.319 | 0.101 | 0.231 | 0.1 | 0.216 | 0.095 | 0.212 | 0.042 | 0.133 | 0.044 | 0.138 | 0.976 | 0.754 | 0.305 | 0.431 | 0.173 | 0.308 | 0.163 | 0.289 | 2.06 | 1.12 | 0.41 | 0.494 | 0.521 | 0.555 |
| 25% | **0.045** | **0.140** | 0.46 | 0.141 | 0.279 | 0.39 | 0.115 | 0.246 | 0.127 | 0.247 | 0.137 | 0.258 | 0.049 | 0.147 | 0.05 | 0.149 | 1.037 | 0.774 | 0.322 | 0.444 | 0.175 | 0.31 | 0.206 | 0.331 | 2.007 | 1.105 | 0.419 | 0.49 | 0.487 | 0.535 |
| 37.50% | **0.049** | **0.147** | 0.052 | 0.151 | 0.4 | 0.465 | 0.126 | 0.257 | 0.158 | 0.276 | 0.187 | 0.304 | 0.056 | 0.158 | 0.06 | 0.163 | 1.107 | 0.8 | 0.353 | 0.462 | 0.185 | 0.315 | 0.252 | 0.37 | 2.033 | 1.111 | 0.429 | 0.498 | 0.487 | 0.529 |
| 50% | **0.056** | **0.158** | 0.06 | 0.162 | 0.602 | 0.572 | 0.136 | 0.268 | 0.183 | 0.299 | 0.232 | 0.341 | 0.065 | 0.17 | 0.068 | 0.173 | 1.193 | 0.838 | 0.369 | 0.472 | 0.212 | 0.339 | 0.316 | 0.419 | 2.054 | 1.119 | 0.467 | 0.529 | 0.484 | 0.523 |
| Avg | **0.048** | **0.144** | 0.049 | 0.146 | 0.367 | 0.436 | 0.119 | 0.25 | 0.142 | 0.259 | 0.163 | 0.279 | 0.053 | 0.152 | 0.055 | 0.156 | 1.079 | 0.792 | 0.337 | 0.452 | 0.186 | 0.318 | 0.234 | 0.352 | 2.039 | 1.114 | 0.431 | 0.503 | 0.495 | 0.475 |
| **Electricity** 12.50% | 0.089 | 0.205 | 0.085 | 0.202 | 0.196 | 0.322 | 0.102 | 0.229 | 0.092 | 0.214 | 0.107 | 0.237 | 0.093 | 0.21 | 0.089 | 0.21 | 0.297 | 0.383 | 0.218 | 0.326 | 0.164 | 0.296 | 0.19 | 0.308 | 0.277 | 0.366 | 0.621 | 0.62 | 0.217 | 0.341 |
| 25% | 0.092 | 0.209 | 0.089 | 0.206 | 0.207 | 0.332 | 0.121 | 0.252 | 0.118 | 0.247 | 0.12 | 0.251 | 0.097 | 0.214 | 0.096 | 0.2 | 0.294 | 0.38 | 0.219 | 0.326 | 0.169 | 0.299 | 0.197 | 0.312 | 0.281 | 0.369 | 0.559 | 0.585 | 0.219 | 0.341 |
| 37.50% | 0.096 | 0.214 | 0.094 | 0.213 | 0.219 | 0.344 | 0.141 | 0.273 | 0.144 | 0.276 | 0.136 | 0.266 | 0.102 | 0.22 | 0.104 | 0.229 | 0.296 | 0.381 | 0.222 | 0.328 | 0.178 | 0.305 | 0.203 | 0.315 | 0.275 | 0.364 | 0.567 | 0.588 | 0.223 | 0.343 |
| 50% | 0.102 | 0.222 | 0.100 | 0.221 | 0.235 | 0.357 | 0.16 | 0.293 | 0.175 | 0.305 | 0.158 | 0.284 | 0.108 | 0.228 | 0.113 | 0.239 | 0.299 | 0.383 | 0.228 | 0.331 | 0.187 | 0.312 | 0.21 | 0.319 | 0.273 | 0.361 | 0.581 | 0.597 | 0.229 | 0.347 |
| Avg | 0.095 | 0.213 | 0.092 | 0.210 | 0.214 | 0.339 | 0.131 | 0.262 | 0.132 | 0.26 | 0.13 | 0.259 | 0.1 | 0.218 | 0.101 | 0.225 | 0.297 | 0.382 | 0.222 | 0.328 | 0.175 | 0.303 | 0.2 | 0.313 | 0.277 | 0.365 | 0.582 | 0.597 | 0.222 | 0.293 |
| **Weather** 12.50% | 0.027 | 0.050 | 0.025 | 0.045 | 0.057 | 0.141 | 0.047 | 0.101 | 0.039 | 0.084 | 0.041 | 0.107 | 0.027 | 0.051 | **0.026** | **0.047** | 0.14 | 0.22 | 0.037 | 0.093 | 0.037 | 0.072 | 0.031 | 0.076 | 0.296 | 0.379 | 0.176 | 0.287 | 0.036 | 0.095 |
| 25% | 0.034 | 0.067 | 0.029 | 0.052 | 0.065 | 0.155 | 0.052 | 0.111 | 0.048 | 0.103 | 0.064 | 0.163 | 0.029 | 0.056 | **0.030** | **0.054** | 0.147 | 0.229 | 0.042 | 0.1 | 0.038 | 0.074 | 0.035 | 0.082 | 0.327 | 0.409 | 0.187 | 0.293 | 0.042 | 0.104 |
| 37.50% | 0.031 | 0.058 | 0.031 | 0.057 | 0.081 | 0.18 | 0.058 | 0.121 | 0.057 | 0.117 | 0.107 | 0.229 | 0.033 | 0.062 | 0.032 | 0.06 | 0.156 | 0.24 | 0.049 | 0.111 | 0.039 | 0.078 | 0.04 | 0.091 | 0.406 | 0.463 | 0.172 | 0.281 | 0.047 | 0.112 |
| 50% | 0.035 | 0.065 | 0.034 | 0.062 | 0.102 | 0.207 | 0.065 | 0.133 | 0.066 | 0.134 | 0.183 | 0.312 | 0.037 | 0.068 | 0.037 | 0.067 | 0.164 | 0.249 | 0.053 | 0.114 | 0.042 | 0.082 | 0.046 | 0.099 | 0.431 | 0.483 | 0.195 | 0.303 | 0.054 | 0.123 |
| Avg | 0.032 | 0.060 | 0.03 | 0.054 | 0.076 | 0.171 | 0.055 | 0.117 | 0.052 | 0.11 | 0.099 | 0.203 | 0.032 | 0.059 | **0.031** | **0.057** | 0.152 | 0.235 | 0.045 | 0.104 | 0.039 | 0.076 | 0.038 | 0.087 | 0.365 | 0.434 | 0.183 | 0.291 | 0.045 | 0.108 |
| Count 1st | 21 | | 32 | | | | | | | | | | | | | | | | | | | | | | | | | | | |

## Table 12: The result of imputation task(order-preserving).

| Models | SpecAR-NET (ours) | | TimesNet (2023) | | ETS. (2022) | | LightTS (2022) | | DLinear (2023) | | FED. (2022) | | Stationary (2022a) | | Auto. (2021) | | Pyra. (2021a) | | In. (2021) | | LogTrans (2019) | | Re. (2020) | | LSTM (1997) | | TCN (2019) | | LSSL (2022) | |
|---|---|---|---|---|---|---|---|---|---|---|---|---|---|---|---|---|---|---|---|---|---|---|---|---|---|---|---|---|---|---|
| MaskRate | MSE | MAE | MSE | MAE | MSE | MAE | MSE | MAE | MSE | MAE | MSE | MAE | MSE | MAE | MSE | MAE | MSE | MAE | MSE | MAE | MSE | MAE | MSE | MAE | MSE | MAE | MSE | MAE | MSE | MAE |
| **ETTm1** 12.50% | **0.018** | **0.089** | 0.019 | 0.092 | 0.067 | 0.188 | 0.075 | 0.18 | 0.058 | 0.162 | 0.035 | 0.135 | 0.026 | 0.107 | 0.034 | 0.124 | 0.67 | 0.541 | 0.047 | 0.155 | 0.041 | 0.141 | 0.032 | 0.126 | 0.974 | 0.78 | 0.51 | 0.493 | 0.101 | 0.231 |
| 25% | **0.022** | **0.098** | 0.023 | 0.101 | 0.096 | 0.229 | 0.093 | 0.206 | 0.08 | 0.193 | 0.052 | 0.166 | 0.032 | 0.119 | 0.046 | 0.144 | 0.689 | 0.553 | 0.063 | 0.18 | 0.044 | 0.144 | 0.042 | 0.146 | 1.032 | 0.807 | 0.518 | 0.5 | 0.106 | 0.235 |
| 37.50% | **0.028** | **0.111** | 0.029 | 0.111 | 0.133 | 0.271 | 0.113 | 0.231 | 0.103 | 0.219 | 0.069 | 0.191 | 0.039 | **0.131** | 0.057 | 0.161 | 0.737 | 0.581 | 0.079 | 0.2 | 0.052 | 0.158 | 0.063 | 0.182 | 0.999 | 0.792 | 0.516 | 0.499 | 0.116 | 0.246 |
| 50% | **0.035** | **0.124** | 0.036 | 0.124 | 0.186 | 0.323 | 0.134 | 0.255 | 0.132 | 0.248 | 0.089 | 0.218 | 0.047 | **0.145** | 0.067 | 0.174 | 0.770 | 0.605 | 0.093 | 0.218 | 0.063 | 0.173 | 0.082 | 0.208 | 0.952 | 0.763 | 0.519 | 0.496 | 0.129 | 0.260 |
| Avg | **0.026** | **0.106** | 0.027 | 0.107 | 0.12 | 0.253 | 0.104 | 0.218 | 0.093 | 0.206 | 0.062 | 0.177 | 0.036 | 0.126 | 0.051 | 0.15 | 0.717 | 0.57 | 0.071 | 0.188 | 0.05 | 0.154 | 0.055 | 0.166 | 0.989 | 0.786 | 0.516 | 0.497 | 0.113 | 0.254 |
| **ETTm2** 12.50% | **0.018** | **0.079** | 0.018 | 0.080 | 0.108 | 0.239 | 0.034 | 0.127 | 0.062 | 0.166 | 0.056 | 0.159 | 0.021 | 0.088 | 0.023 | 0.092 | 0.394 | 0.470 | 0.133 | 0.27 | 0.103 | 0.229 | 0.108 | 0.228 | 1.013 | 0.805 | 0.307 | 0.441 | 0.150 | 0.298 |
| 25% | **0.020** | **0.084** | 0.020 | 0.085 | 0.164 | 0.294 | 0.042 | 0.143 | 0.085 | 0.196 | 0.080 | 0.195 | 0.024 | 0.096 | 0.026 | 0.101 | 0.421 | 0.482 | 0.135 | 0.272 | 0.120 | 0.248 | 0.136 | 0.262 | 1.039 | 0.814 | 0.263 | 0.402 | 0.159 | 0.306 |
| 37.50% | **0.022** | **0.089** | 0.023 | 0.091 | 0.237 | 0.356 | 0.051 | 0.159 | 0.106 | 0.222 | 0.11 | 0.231 | 0.027 | 0.103 | 0.03 | 0.108 | 0.478 | 0.521 | 0.155 | 0.293 | 0.138 | 0.26 | 0.175 | 0.3 | 0.917 | 0.744 | 0.25 | 0.396 | 0.18 | 0.321 |
| 50% | **0.025** | **0.097** | 0.026 | 0.098 | 0.323 | 0.421 | 0.059 | 0.174 | 0.131 | 0.247 | 0.156 | 0.276 | 0.03 | 0.108 | 0.035 | 0.119 | 0.568 | 0.56 | 0.2 | 0.333 | 0.117 | 0.247 | 0.211 | 0.329 | 1.14 | 0.835 | 0.246 | 0.389 | 0.21 | 0.353 |
| Avg | **0.021** | **0.087** | 0.022 | 0.089 | 0.208 | 0.327 | 0.046 | 0.151 | 0.096 | 0.208 | 0.101 | 0.215 | 0.026 | 0.099 | 0.029 | 0.11 | 0.465 | 0.508 | 0.156 | 0.292 | 0.119 | 0.246 | 0.157 | 0.28 | 1.027 | 0.8 | 0.266 | 0.407 | 0.175 | 0.324 |
| **ETTh1** 12.50% | **0.044** | **0.144** | 0.057 | 0.159 | 0.126 | 0.263 | 0.24 | 0.345 | 0.151 | 0.267 | 0.07 | 0.19 | 0.06 | 0.165 | 0.074 | 0.182 | 0.857 | 0.609 | 0.114 | 0.234 | 0.229 | 0.33 | 0.074 | 0.194 | 1.265 | 0.896 | 0.599 | 0.554 | 0.422 | 0.461 |
| 25% | **0.061** | **0.169** | 0.069 | 0.178 | 0.169 | 0.304 | 0.265 | 0.364 | 0.18 | 0.292 | 0.106 | 0.236 | 0.08 | 0.189 | 0.09 | 0.203 | 0.829 | 0.672 | 0.14 | 0.262 | 0.207 | 0.323 | 0.102 | 0.227 | 1.262 | 0.883 | 0.61 | 0.567 | 0.412 | 0.456 |
| 37.50% | **0.079** | **0.190** | 0.084 | 0.196 | 0.22 | 0.347 | 0.296 | 0.382 | 0.215 | 0.318 | 0.124 | 0.258 | 0.102 | 0.212 | 0.109 | 0.222 | 0.83 | 0.675 | 0.174 | 0.293 | 0.21 | 0.328 | 0.135 | 0.261 | 1.2 | 0.867 | 0.628 | 0.577 | 0.421 | 0.461 |
| 50% | **0.098** | **0.210** | 0.102 | 0.215 | 0.293 | 0.402 | 0.334 | 0.404 | 0.257 | 0.347 | 0.165 | 0.299 | 0.133 | 0.24 | 0.137 | 0.248 | 0.854 | 0.691 | 0.215 | 0.325 | 0.23 | 0.348 | 0.179 | 0.298 | 1.174 | 0.849 | 0.648 | 0.587 | 0.443 | 0.473 |
| Avg | **0.0705** | **0.17825** | 0.078 | 0.187 | 0.202 | 0.329 | 0.284 | 0.373 | 0.201 | 0.306 | 0.117 | 0.246 | 0.094 | 0.201 | 0.103 | 0.214 | 0.842 | 0.682 | 0.161 | 0.279 | 0.219 | 0.332 | 0.122 | 0.245 | 1.225 | 0.873 | 0.621 | 0.571 | 0.424 | 0.481 |
| **ETTh2** 12.50% | **0.038** | **0.128** | 0.04 | 0.13 | 0.187 | 0.319 | 0.101 | 0.231 | 0.1 | 0.216 | 0.095 | 0.212 | 0.042 | 0.133 | 0.044 | 0.138 | 0.976 | 0.754 | 0.305 | 0.431 | 0.173 | 0.308 | 0.163 | 0.289 | 2.06 | 1.12 | 0.41 | 0.494 | 0.521 | 0.555 |
| 25% | **0.042** | **0.136** | 0.46 | 0.141 | 0.279 | 0.39 | 0.115 | 0.246 | 0.127 | 0.247 | 0.137 | 0.258 | 0.049 | 0.147 | 0.05 | 0.149 | 1.037 | 0.774 | 0.322 | 0.444 | 0.175 | 0.31 | 0.206 | 0.331 | 2.007 | 1.105 | 0.419 | 0.49 | 0.487 | 0.535 |
| 37.50% | **0.047** | **0.144** | 0.052 | 0.151 | 0.4 | 0.465 | 0.126 | 0.257 | 0.158 | 0.276 | 0.187 | 0.304 | 0.056 | 0.158 | 0.06 | 0.163 | 1.107 | 0.8 | 0.353 | 0.462 | 0.185 | 0.315 | 0.252 | 0.37 | 2.033 | 1.111 | 0.429 | 0.498 | 0.487 | 0.529 |
| 50% | **0.056** | **0.157** | 0.06 | 0.162 | 0.602 | 0.572 | 0.136 | 0.268 | 0.183 | 0.299 | 0.232 | 0.341 | 0.065 | 0.17 | 0.068 | 0.173 | 1.193 | 0.838 | 0.369 | 0.472 | 0.212 | 0.339 | 0.316 | 0.419 | 2.054 | 1.119 | 0.467 | 0.529 | 0.484 | 0.523 |
| Avg | **0.046** | **0.141** | 0.049 | 0.146 | 0.367 | 0.436 | 0.119 | 0.25 | 0.142 | 0.259 | 0.163 | 0.279 | 0.053 | 0.152 | 0.055 | 0.156 | 1.079 | 0.792 | 0.337 | 0.452 | 0.186 | 0.318 | 0.234 | 0.352 | 2.039 | 1.114 | 0.431 | 0.503 | 0.495 | 0.475 |
| **Electricity** 12.50% | **0.086** | **0.202** | 0.085 | 0.202 | 0.196 | 0.321 | 0.102 | 0.229 | 0.092 | 0.214 | 0.107 | 0.237 | 0.093 | **0.210** | 0.089 | 0.21 | 0.297 | 0.383 | 0.218 | 0.326 | 0.164 | 0.296 | 0.19 | 0.308 | 0.277 | 0.366 | 0.621 | 0.62 | 0.217 | 0.341 |
| 25% | **0.089** | **0.206** | 0.089 | 0.206 | 0.207 | 0.332 | 0.121 | 0.252 | 0.118 | 0.247 | 0.12 | 0.251 | 0.097 | **0.214** | 0.096 | 0.22 | 0.294 | 0.38 | 0.219 | 0.326 | 0.169 | 0.299 | 0.197 | 0.312 | 0.281 | 0.369 | 0.559 | 0.585 | 0.219 | 0.341 |
| 37.50% | **0.094** | **0.212** | 0.094 | 0.213 | 0.219 | 0.344 | 0.141 | 0.273 | 0.144 | 0.276 | 0.136 | 0.266 | **0.102** | 0.22 | 0.104 | 0.229 | 0.296 | 0.381 | 0.222 | 0.328 | 0.178 | 0.305 | 0.203 | 0.315 | 0.275 | 0.364 | 0.567 | 0.588 | 0.223 | 0.343 |
| 50% | **0.100** | **0.220** | 0.100 | 0.221 | 0.235 | 0.357 | 0.16 | 0.293 | 0.175 | 0.305 | 0.158 | 0.284 | **0.108** | 0.228 | 0.113 | 0.239 | 0.299 | 0.383 | 0.228 | 0.331 | 0.187 | 0.312 | 0.21 | 0.319 | 0.273 | 0.361 | 0.581 | 0.597 | 0.229 | 0.347 |
| Avg | **0.092** | **0.210** | 0.092 | 0.210 | 0.214 | 0.339 | 0.131 | 0.262 | 0.132 | 0.26 | 0.13 | 0.259 | **0.100** | **0.218** | 0.101 | 0.225 | 0.297 | 0.382 | 0.222 | 0.328 | 0.175 | 0.303 | 0.2 | 0.313 | 0.277 | 0.365 | 0.582 | 0.597 | 0.222 | 0.293 |
| **Weather** 12.50% | 0.027 | 0.052 | 0.025 | 0.045 | 0.057 | 0.141 | 0.047 | 0.101 | 0.039 | 0.084 | 0.041 | 0.107 | 0.027 | 0.051 | **0.026** | **0.047** | 0.14 | 0.22 | 0.037 | 0.093 | 0.037 | 0.072 | 0.031 | 0.076 | 0.296 | 0.379 | 0.176 | 0.287 | 0.036 | 0.095 |
| 25% | **0.028** | **0.052** | 0.029 | 0.052 | 0.065 | 0.155 | 0.052 | 0.111 | 0.048 | 0.103 | 0.064 | 0.163 | 0.029 | 0.056 | 0.030 | 0.054 | 0.147 | 0.229 | 0.042 | 0.1 | 0.038 | 0.074 | 0.035 | 0.082 | 0.327 | 0.409 | 0.187 | 0.293 | 0.042 | 0.104 |
| 37.50% | **0.031** | **0.058** | 0.031 | 0.057 | 0.081 | 0.18 | 0.058 | 0.121 | 0.057 | 0.117 | 0.107 | 0.229 | 0.033 | 0.062 | 0.032 | 0.06 | 0.156 | 0.24 | 0.049 | 0.111 | 0.039 | 0.078 | 0.04 | 0.091 | 0.406 | 0.463 | 0.172 | 0.281 | 0.047 | 0.112 |
| 50% | **0.036** | **0.066** | 0.034 | 0.062 | 0.102 | 0.207 | 0.065 | 0.133 | 0.066 | 0.134 | 0.183 | 0.312 | 0.037 | 0.068 | 0.037 | 0.067 | 0.164 | 0.249 | 0.053 | 0.114 | 0.042 | 0.082 | 0.046 | 0.099 | 0.431 | 0.483 | 0.195 | 0.303 | 0.054 | 0.123 |
| Avg | **0.031** | **0.057** | 0.03 | 0.054 | 0.076 | 0.171 | 0.055 | 0.117 | 0.052 | 0.11 | 0.099 | 0.203 | 0.032 | 0.059 | **0.031** | **0.057** | 0.152 | 0.235 | 0.045 | 0.104 | 0.039 | 0.076 | 0.038 | 0.087 | 0.365 | 0.434 | 0.183 | 0.291 | 0.045 | 0.108 |
| Count 1st | 42 | | 17 | | | | | | | | | | | | | | | | | | | | | | | | | | | |