# OpenReview forum: "SpecAR-Net: Spectrogram Analysis and Representation Network for Time Series"
_ICLR.cc/2024/Conference — Submitted to ICLR 2024_

### Official Review · Reviewer_13Yj · 2023-10-30

**Soundness:** 1 poor
**Presentation:** 2 fair
**Contribution:** 2 fair
**Rating:** 3
**Confidence:** 3

**Summary:**

A new model for time-series analysis, SpecAR-Net, is proposed. SpecAR-Net works in the time-frequency domain, and it outperforms existing methods in various time series analysis tasks, including classification, anomaly detection, imputation, and long- and short-term series forecasting.

**Strengths:**

- SpecAR-Block can be used in a plug-in manner.

- The experiment includes many SOTA models for comparison.

**Weaknesses:**

- Unifying time and frequency domains is not a novel idea (e.g., [Woo et al., 2022]).

- In Introduction,
> i. For decoupling the periodicity characteristics from time series data, the time-frequency transform is used for better extraction of time-frequency variation patterns in a higher dimensional feature space.
> ii. The mutagenicity disrupt the stability of the semantic representation space for time series.
To address such issue, a group of parallel multi-scale convolution blocks is designed to
deeply explore the transient patterns.
> iii. To capture the trend patterns, the order-preserving is added to the loss function. This learning
strategy, guided by the It is not clear that the proposed methods address the corresponding problems respectively.
It is not clear that the proposed methods address the corresponding problems.

- The code for reproducibility is not open.

- It is unclear if the experimental results are statistically significant, and thus they are not convincing.

- Experimental setting is not clearly stated.

**Questions:**

- [Comment] All figures should be vector images.

- [Question (major)] It improves the paper quality to include ablation studies about periodicity, mutagenicity, and trend patterns. Do your proposed methods (time-frequency transform, multi-scale convolution, and order-preserving, respectively) address these problems?

- [Comment] In Introduction,
> The mutagenicity disrupt the stability of the semantic representation space for time series. To address such issue, a group of parallel multi-scale convolution blocks is designed to deeply explore the transient patterns.
The multi-scale convolution is not a totally new idea, as is mentioned in Related Work, and I would like to recommend that the authors cite relevant reference papers here.

- [Question] Are the number of parameters of the models used in the experiment comparable? Is the comparison fair?

- [Question (major)] What is the whole search space of hyperparameters?

- [Question (major)] How did you tune the hyperparameters listed in the paper? How about ones not listed in the paper?

- [Comment (major)] I strongly recommend submitting the code to reproduce the results.

- [Typo] In Table 3, 2..051 should be 2.051.

---

### Official Review · Reviewer_forZ · 2023-10-31

**Soundness:** 2 fair
**Presentation:** 2 fair
**Contribution:** 2 fair
**Rating:** 3
**Confidence:** 4

**Summary:**

This paper presents Spectrum Analysis and Representation Network (SpecAR-Net). SpecAR-Net aims at learning more comprehensive representations by modeling raw time series in both time and frequency domain, where an efficient joint extraction of time-frequency features is achieved through a group of learnable 2D multi-scale parallel complex convolution blocks. SpecAR-Net achieves excellent performance in 5 time series tasks i.e., classification, anomaly detection, imputation, long- and short-term series forecasting.

**Strengths:**

1.	Various complex convolutions in the frequency domain is proposed for time series forecasting.
2.	The experiments shows that the proposed SpecAR-Net framework outperforms most SOTA bases in 5 downstream tasks.

**Weaknesses:**

1.	It is claimed that the proposed SpecAR-Net is a plug-and-play time series representation module. Unfortunately, this claimed is fully supported. For example, the plugging of the proposed structure in other models is not studied and reported.
2.	As the frequency domain based methods have been applied widely in time series, such as TimesNet and Fedformer. In fact, the proposed framework is very similar to TimesNet.  It seems that the primary distinction between proposed methods and TimeNnet is the modification of convolutions into the frequency domain. Are there any other differences that contribute to the advancements in downstream tasks?
3.	More analysis experiments are welcome to make the empirical studies convincing. For example, I would recommend add the experiment with Wavelet transform. What if the STFT is replaced by Wavelet Transform? It would be beneficial to add such experiments.
4.	The writing can be further improved. For example, the first several sentences of Abstract are not directly related to the contribution of this paper. The resolution of Figure 1 and Figure 2 is too low to check the details.

**Questions:**

In the TF Resolution experiment, it is observed that a larger window length proves advantageous for predicting lengths of 336 and 720 and only marginal improvement for predicting lengths of 96 and 192. The reason why a larger window length fails to provide significant benefits for shorter prediction lengths is unclear. Please explain.

---

### Official Review · Reviewer_hAhz · 2023-11-01

**Soundness:** 2 fair
**Presentation:** 2 fair
**Contribution:** 1 poor
**Rating:** 5
**Confidence:** 3

**Summary:**

The paper introduces SpecAR-Net (Spectrum Analysis and Representation Network) for representing 1D time-series data. SpecAR-Net consists of SpecAR-Blocks, where each block transforms 1D times-series data into a 2D time-frequency spectrogram via short-term Fourier transform  (STFT) followed by multi-scale 2D convolutions and pooling. Experimental results on five downstream tasks demonstrate that SpecAR-Net is competitive relative to baselines.

**Strengths:**

- The proposed framework is quite general and could be incorporated into several frameworks (CNNs, RNNs, or Transformers)
- SpecAR-Net is competitive relative to baselines on several benchmark tasks
- Time-frequency modeling of time series data enables the capturing of time-dependent trends and periodic components.

**Weaknesses:**

- SpecAR-Net seems like a straightforward combination of STFT and multi-scale convolutions for time-series data. Limited insights/justifications are provided for the assumed modeling choices.
- The proposed approach consists of multiple complex components (short-term Fourier transform  (STFT) followed by multi-scale 2D convolutions, pooling, and temporal order preserving constraint). The paper does not provide justification for model choices, e.g.,
1) STFT vs. Wavelet transforms:  Unlike the STFT which uses a fixed size window, the Wavelet Transform uses a variable size window to analyze the signal, allowing for better time resolution at high frequencies and better frequency resolution at low frequencies.
2) While the paper opts for the Hamming window, alternative windows could be considered, e.g.,  rectangular window (no windowing), Hann window,  Blackman window, Gaussian window, etc. Each of these has different properties and is suitable for different situations. It's often beneficial to experiment with different windows to see which one works best for a specific application.
- Several works jointly modeling time and frequencies of 1D time-series have been proposed, it's unclear what differentiates SpecAR-net from such approaches, e.g., TimesNet.


**Minor**
- The font size of the following figures should be increased for clarity: Figures 1, 3

**Questions:**

- Could you provide an ablation study on all the SpecAR-Net components?
- It seems the performance of TimesNet is comparable to SpecAR-Net.
1) What are the limitations/benefits of using either approach?
2) What are the performance error bars in Tables 3-7?

---

### Meta-Review · Area_Chair_7v9n · 2023-12-06

**Metareview:**

This paper proposes the use of spectrograms in time series analysis. The reviewers pointed out issues with the novelty of the work and deemed the paper a straightforward combination of past work. The differentiation compared to prior work is also unclear. The reviewers had multiple other questions about the experiments. The authors have not responded to the reviewers.

**Justification For Why Not Higher Score:**

There are no grounds to overturn reviewer consensus.

**Justification For Why Not Lower Score:**

N/A

---

### Decision · Program_Chairs · 2024-01-16

Reject